# Off-targets of BRAF inhibitors disrupt endothelial signaling and vascular barrier function

Sophie Bromberger[1], Yuliia Zadorozhna[1], Julia Maria Ressler[1], Silvio Holzner[1], Arkadiusz Nawrocki[2], Nina Zila[1,3], Alexander Springer[4], Martin Røssel Larsen[2], Klaudia Schossleitner[1]

Targeted therapies against mutant BRAF are effectively used in combination with MEK inhibitors (MEKi) to treat advanced melanoma. However, treatment success is affected by resistance and adverse events (AEs). Approved BRAF inhibitors (BRAFi) show high levels of target promiscuity, which can contribute to these effects. The blood vessel lining is in direct contact with high plasma concentrations of BRAFi, but effects of the inhibitors in this cell type are unknown. Hence, we aimed to characterize responses to approved BRAFi for melanoma in the vascular endothelium. We showed that clinically approved BRAFi induced a paradoxical activation of endothelial MAPK signaling. Moreover, phosphoproteomics revealed distinct sets of off-targets per inhibitor. Endothelial barrier function and junction integrity were impaired upon treatment with vemurafenib and the next-generation dimerization inhibitor PLX8394, but not with dabrafenib or encorafenib. Together, these findings provide insights into the surprisingly distinct side effects of BRAFi on endothelial signaling and functionality. Better understanding of off-target effects could help to identify molecular mechanisms behind AEs and guide the continued development of therapies for BRAF-mutant melanoma.

## Introduction

Melanoma is a highly aggressive form of skin cancer and is associated with a high mortality rate (Schadendorf et al, 2018). According to previous literature, ~50% of melanomas harbor mutations in the *BRAF* gene, of which the vast majority encodes the BRAF-V600E oncoprotein (Davies et al, 2002; Ascierto et al, 2012). This mutation induces the constitutive activation of BRAF and downstream MAPK signaling and subsequently promotes excessive proliferation and survival of tumor cells. Targeted therapies, including inhibitors of mutant BRAF or its downstream effector MEK, are used to suppress this pathway in patients, with a combined

approach yielding the best outcomes (Flaherty et al, 2012; Ascierto et al, 2016). Currently, three BRAF inhibitors (BRAFi) are clinically approved for the treatment of BRAF-V600E and BRAF-V600K mutant melanoma, namely vemurafenib, dabrafenib, and encorafenib, commonly administered together with the MEK inhibitors (MEKi) cobimentinib, trametinib, or binimetinib, respectively (Larkin et al, 2014; Long et al, 2015; Dummer et al, 2018). Although these targeted therapies have greatly improved the prognosis of patients with advanced BRAF-mutant melanoma, they also have two major limitations: on the one hand, acquired resistance to BRAF inhibition typically develops after a median of 9–12 mo (Larkin et al, 2014; Long et al, 2015; Dummer et al, 2018). On the other hand, patients often experience adverse events (AEs), which lead to a discontinuation rate of up to 15.7% and to dose modifications in about 50% of patients (Heinzerling et al, 2019).

Numerous molecular processes potentially causing a resistance to BRAF inhibition have been studied and reviewed, among them MAPK-dependent and -independent mechanisms (Holderfield et al, 2014; Luebker & Koepsell, 2019). Yet the underlying molecular mechanisms for AEs remain largely unknown. It is often proposed that resistance mechanisms and AEs can arise from a phenomenon called paradoxical ERK activation, which describes an activation of downstream MAPK signaling upon BRAF inhibition (Poulikakos et al, 2010; Adelmann et al, 2016). This phenomenon is caused by an alteration of RAS-dependent dimerization of BRAF (Lavoie et al, 2013). Newer drug development strategies include so-called "paradox breakers", which are dimerization inhibitors designed to avoid paradoxical ERK activation (Brummer & McInnes, 2020). However, this inhibitor class still has to be clinically evaluated.

Furthermore, paradoxical activation of the MAPK pathway is not the only mechanism responsible for resistance and AEs. Therapeutic kinase inhibitors have repeatedly been investigated for their polypharmacology, meaning that they have binding capacities for a number of proteins aside from their designated target. For example, vemurafenib has been shown to inhibit not only mutant BRAF but also WT BRAF and CRAF in cell-free assays (Bollag et al, 2010). In higher concentrations it can inhibit a variety of other kinases,

[1]Department of Dermatology, Medical University of Vienna, Vienna, Austria   [2]Department of Biochemistry and Molecular Biology, University of Southern Denmark, Odense, Denmark   [3]University of Applied Sciences FH Campus Wien, Division of Biomedical Science, Vienna, Austria   [4]Department of Pediatric Surgery, Medical University of Vienna, Vienna, Austria

Correspondence: klaudia.schossleitner@meduniwien.ac.at

including LCK, YES1, SRC, or CSK. Dabrafenib has also been reported to act on WT BRAF and CRAF (Rheault et al, 2013). A comprehensive investigation on the target promiscuity of these inhibitors has been published in 2017, elucidating their binding capacities in protein lysates of cancer cells (Klaeger et al, 2017).

In recent years, it has been increasingly recognized that BRAFi have off-target effects on stromal cells of the tumor microenvironment (TME), such as fibroblasts, but also on immune cells, and that these off-targets could have a crucial impact on the treatment outcome (Callahan et al, 2014; Corrales et al, 2021; Loria et al, 2022). However, the vascular system has been severely underrepresented in this line of research, even though the vascular endothelium is in contact with high concentrations of BRAFi in the circulation. For example, patients receiving vemurafenib experience plasma levels of up to 61.4 $\mu g/ml$ ($\hat{=}$ 125.32 $\mu M$), which easily exceeds thresholds for interactions with multiple off-target kinases (Bollag et al, 2010; European Medicines Agency, 2012).

Impaired vascular function can be problematic, especially for patients with comorbidities (Mincu et al, 2019; Lyon et al, 2020). For example, endothelial dysfunction reduces the ability of blood vessels to dilate and can lead to increased peripheral resistance, a hallmark of hypertension (Vanhoutte et al, 2017; Ma et al, 2023). The endothelium helps to regulate the balance between pro- and anticoagulant mechanisms, and vascular damage can be a cause for disproportionate coagulation events, including thrombosis or hemorrhage (Neubauer & Zieger, 2022). Activation of adhesion receptors on endothelial cells can contribute to protumorigenic immune cell infiltrates and the formation of metastatic niches (Reymond et al, 2013; Häuselmann et al, 2016; Wettschureck et al, 2019). Increased permeability and the subsequent accumulation of excess fluid leads to higher interstitial pressure, which can limit treatment perfusion of the tumor and consequently can reduce therapeutic efficacy (Goel et al, 2011). No treatment against endothelial activation and vascular barrier disruption is available to date (Claesson-Welsh et al, 2021). Thus, in depth knowledge of the molecular signaling mechanisms in human endothelial cells is needed to inform future studies and therapeutic development.

In the present study, we aimed at elucidating the effects of BRAFi treatment on vascular endothelial signaling and functionality. We observed that paradoxical ERK activation occurs in endothelial cells. Simultaneously, numerous other signaling cascades were affected by BRAFi treatment, which we could show in a global mass spectrometry (MS)-based phosphoproteomics analysis. The comparison of several clinically used BRAFi revealed that endothelial off-targets were highly variable among treatments. Essential endothelial functions, most prominently the endothelial barrier, were also differentially affected by BRAFi treatment. Together, our data provide insights into the mechanisms of BRAFi-induced endothelial signaling disruption and dysfunction, which adds another piece to the puzzle of understanding the role of the TME in treatment outcomes and AEs in advanced melanoma.

# Results

### BRAFi induce paradoxical MAPK signaling in endothelial cells

Vemurafenib, dabrafenib, encorafenib and the next-generation dimerization inhibitor PLX8394 were designed to specifically

target BRAF bearing the V600E mutation. However, to a certain extent, these inhibitors also act on WT BRAF and other off-targets in tumor cells in a concentration-dependent manner (Bollag et al, 2010; Rheault et al, 2013; Klaeger et al, 2017). In Fig 1A, we could indeed show that increasing concentrations of vemurafenib inhibited downstream ERK1/2 phosphorylation (T202/Y204) in primary BRAF-mutant melanoma cells and to a lesser degree in BRAF-WT melanoma cells after 1 h. In contrast, NRAS-mutant melanoma cells displayed a paradoxical activation of pERK1/2 after treatment with 1–10 $\mu M$ vemurafenib, which is in line with previous reports (Oh et al, 2016). Notably, when dermal micro-vascular endothelial cells (DMEC) were stimulated with the same concentrations of BRAFi, these cells also showed elevated pERK1/2 levels (Fig 1B and C). Paradoxical activation could be seen after treatment of DMEC with low doses (1 $\mu M$) of dabrafenib and encorafenib (Fig 1D). Interestingly, the concentration at which we saw activation of ERK1/2 after dabrafenib and encorafenib was similar, but the peak of vemurafenib-induced activation occurred at a higher concentration of 10 $\mu M$ (Fig 1D). To investigate if the phosphorylation pattern in endothelial cells follows what is known for paradoxical activation of BRAF in melanoma cells, we added PLX8394, an inhibitor designed to avoid paradoxical activation, and indeed PLX8394-treated samples did not show elevated pERK levels in either cell type. All BRAFi were added at concentrations relevant for human use. Whereas vemurafenib had a maximum plasma concentration ($C_{max}$) of up to 125.32 $\mu M$ in clinical studies, standard dosing of dabrafenib and encorafenib led to plasma levels of 2.84 and 11 $\mu M$, respectively (European Medicines Agency, 2012, 2013, 2018). Although all BRAFi efficiently impeded ERK phosphorylation in BRAF-mutant and BRAF-WT melanoma cells, their effect on MAPK signaling in endothelial cells resembled the effect in melanoma cells with upstream NRAS mutations (Fig S1).

### Phosphoproteomics reveals BRAFi-induced disruption of endothelial signaling

To investigate if clinically relevant concentrations of BRAFi not only induce direct effects on MAPK signaling but also induce off-target effects in endothelial cells, we used an MS-based phosphoproteomics approach to determine altered phosphosites in phosphoproteins in DMEC after 1 h of BRAFi treatment. Whereas BRAFi did not affect overall protein expression after 1 h (Fig S2), our analysis of phosphopeptides in Fig 2A revealed distinct phosphorylation patterns for the tested inhibitors. The striking heterogeneity in phosphorylation among treatments became even more obvious when we found that most of the significantly altered phosphosites were unique for the individual inhibitors (Fig 2B). In more detail, only two phosphosites were commonly inhibited by all treatments, namely Desmoplakin (DSP, S2209) and Band 4.1-like protein 2 (EPB41L2, S87). A decrease in phosphorylation of Cortactin (CTTN, S261), Paxillin (PXN, S270), RHO GTPase-activating protein 29 (ARHGAP29, S356), and Liprin-beta-1 (PPFIBP1, S908) was observed in all treatments except 10 $\mu M$ of vemurafenib. We observed that dabrafenib (10 $\mu M$) treatment led to the highest number of altered phosphosites, namely 107. Interestingly, the same concentration of vemurafenib induced only minimal changes, whereas the higher concentration had stronger effects with 4 versus 95 significantly

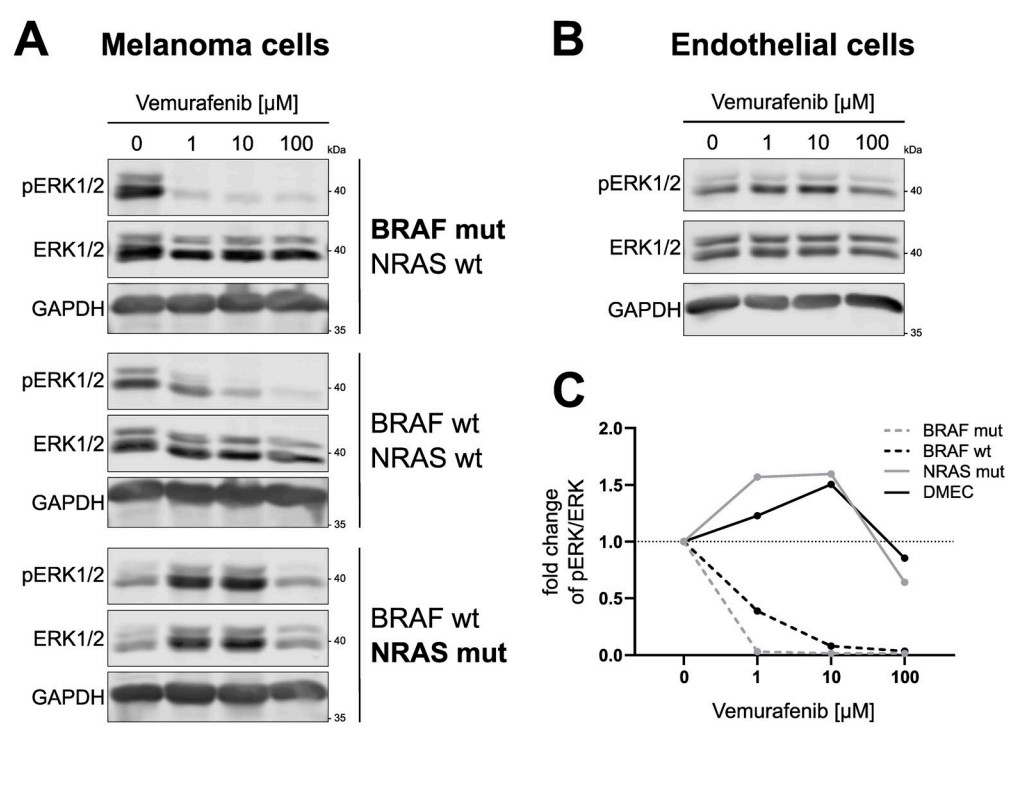

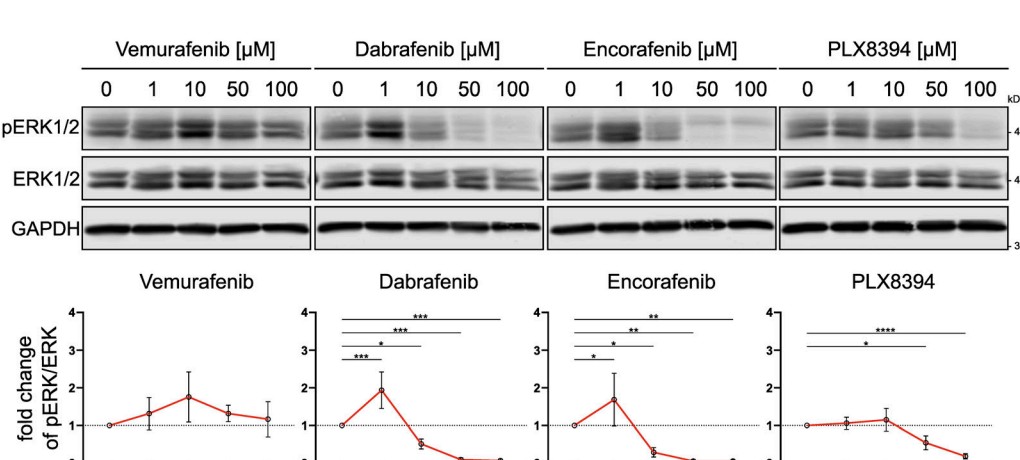

**Figure 1. BRAFi induce paradoxical ERK1/2 activation in endothelial cells.**
**(A, B)** Fluorescence-based detection of pERK1/2 (T202/Y204), total ERK1/2 and GAPDH in Western blots of melanoma cells (A) and dermal microvascular endothelial cells (B), treated with indicated concentrations of vemurafenib for 1 h. **(C)** Quantification of band intensities of blots shown in A+B (pERK/ERK ratio) displayed as fold change from the respective vehicle control. **(D)** Fluorescence-based detection and respective quantifications of ERK1/2 phosphorylation in Western blots of dermal microvascular endothelial cells treated with indicated BRAFi concentrations for 1 h. Quantification of band intensities is displayed as fold change of pERK/ERK ratio from the vehicle control (mean ± SD, n = 3–4).

altered phosphosites (Fig 2C). Reactome pathway enrichment analysis of the differentially phosphorylated proteins revealed that each of the inhibitors induced changes in its individual set of pathways (Fig 2D), which again highlighted the different global effects of BRAFi. For example, vemurafenib (100 µM) interfered with pathways involved in mTOR and RHO GTPase signaling, whereas the

effects of dabrafenib were associated with different aspects of the MAPK pathway.

We then integrated our experimental phosphopeptide abundance data with known kinase-substrate interactions to predict kinase activity via the previously published KinSwingR package (Engholm-Keller et al, 2019). Based on the phosphosites present in

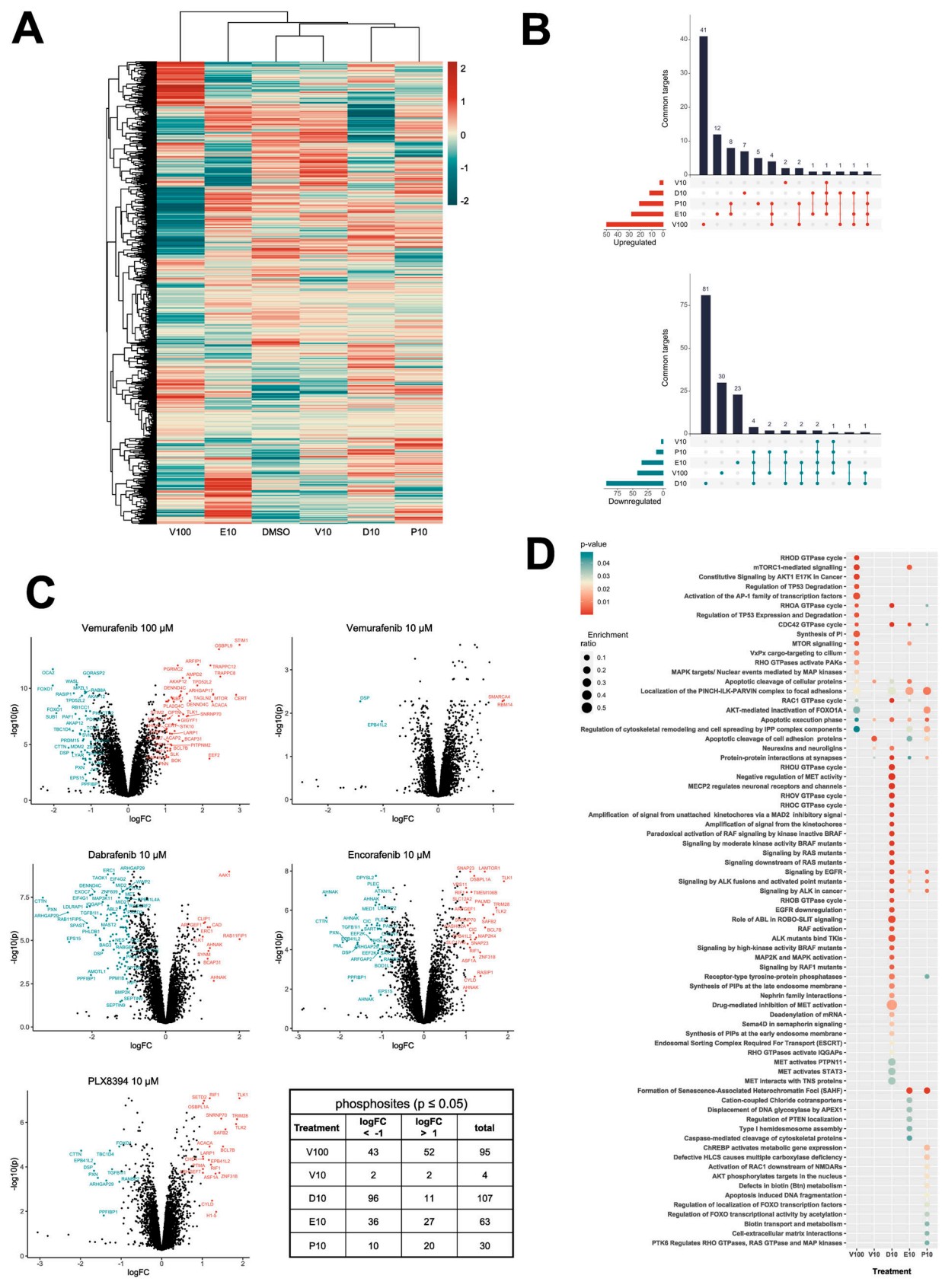

our dataset, we identified potential substrates for 156 kinases and computed their activity scores for each inhibitor treatment compared with the control. Hierarchical clustering of the 50 most differentially active kinases among inhibitor treatments emphasized the sometimes-contrasting effects of the used BRAFi on the vasculature (Fig 3A). STRING-based physical interaction networks of those kinases revealed that several CDKs and MAPKs, as well as GSK3-$\alpha/\beta$, were inhibited with dabrafenib and the high dose of vemurafenib, whereas Src-family kinases, AKT1, and protein kinases A and C were particularly activated with dabrafenib (Fig 3B). Encorafenib only had mild effects on predicted kinase activity, despite manipulating multiple phosphoproteins. Klaeger et al published an extensive study using cancer cell lysates to investigate the target promiscuity of 243 clinical kinase inhibitors, including vemurafenib, dabrafenib, and encorafenib (Klaeger et al, 2017). They used a competitive affinity assay with immobilized broad-spectrum inhibitors (kinobeads) combined with mass spectrometry-based protein quantification, to assess which proteins would be bound by individual kinase inhibitors in lysates from leukemia, neuroblastoma, and adenocarcinoma cell lines. We compared their datasets with the kinase activity predictions in our data from endothelial cells to deduce which off-targets could be directly bound by BRAFi and which could be downstream effectors (Table 1). Klaeger et al (2017) identified 10 proteins that were directly bound by vemurafenib (up to 30 $\mu$M), three of which also occurred in our kinase dataset (BRAF, PTK6, TGFBR2). Notably, BRAF was paradoxically activated by 10 $\mu$M but inhibited by 100 $\mu$M vemurafenib. We found an overlap of 24 kinases that were directly bound by dabrafenib and were also present in our prediction dataset (total of 56 proteins in the Klaeger dataset). Especially a group of CDKs were strongly inhibited by dabrafenib and also shown to be physically bound in low $\mu$M concentrations. Other kinases that were present in both datasets include ABL1/2, HCK, JAK2, and YES1. Interestingly, some kinases that were identified in the Klaeger dataset (CAMK4, FYN, and LCK) were activated by dabrafenib treatment. Encorafenib bound 28 proteins in the Klaeger dataset, 10 of which we also identified. In this case, especially GSK3-$\alpha/\beta$ as well as MAPK8/9 and MAPKAPK2 were inhibited in encorafenib treated samples by direct interaction. Our phosphoproteomics and kinase prediction data clearly highlight different off-targets among clinically used BRAFi in endothelial cells, even though these molecules were all designed to target mutant BRAF in melanoma cells.

### BRAFi differentially affect endothelial barrier function

Next, we investigated if the diverse effects on signaling pathways have functional consequences in endothelial cells. After 1 h of exposure, endothelial cell viability was not affected significantly by any inhibitor except for dabrafenib, with viability remaining above 90% for all tested concentrations (Fig S3). Furthermore, BRAFi did

not induce surface expression of activation markers such as ICAM-1 and E-Selectin (Fig S4). However, electrical cell-substrate impedance sensing (ECIS) measurements of DMEC monolayers revealed a substantial dose-dependent disruption of electrical barrier resistance (measured at 250 Hz) by vemurafenib (Fig 4A). After 1 h, vemurafenib induced a significant decrease in barrier resistance at concentrations ranging from 10 to 100 $\mu$M. Dabrafenib disrupted the endothelial barrier only at the highest concentration, whereas encorafenib had no significant effect, even at high concentrations. Surprisingly, PLX8394 treatment also induced a drop in barrier resistance at 50 and 100 $\mu$M. The impact of BRAFi on electrical barrier resistance was similar between blood endothelial cells (BEC) and lymphatic endothelial cells (LEC, Fig S5). Concurrently, high doses of vemurafenib and PLX8394 increased endothelial permeability of high (70 kD) and low (376 D) molecular weight tracers in a transwell assay after 1 and 6 h when compared with the vehicle control (Fig 4B). Dabrafenib and encorafenib had no effect on tracer permeability in this assay. In addition, we observed that high doses of vemurafenib induced visible disruptions of endothelial cell-cell junctions (Fig 5A). Tight and adherens junctions appeared smooth and continuous in vehicle control-treated DMEC monolayers, as visualized by immunofluorescence of claudin-5 and vascular endothelial (VE)-cadherin. In contrast, junctions were visibly interrupted and disorganized upon treatment with 100 $\mu$M of vemurafenib, and the junction fragments appeared attached to actin stress fibers. Junction disruption also manifested as decreased fluorescence intensity of junctional markers claudin-5 and VE-cadherin. Dabrafenib did not alter the junction architecture to the same extent (Figs 5A and S6). Of note, 100 $\mu$M of PLX8394 also induced visible junction disruptions and intercellular gaps (Fig S7). BEC and LEC were differentiated by Prox1 staining and showed comparable effects on junction architecture upon BRAFi treatment (Figs S6 and S7).

A weak endothelial barrier can have detrimental consequences, not only because of the leakage of fluid and small molecules, but also during metastasis formation. Based on a previously published in vitro model of tumor cell invasion (Holzner et al, 2016), we measured the size of melanoma spheroid-induced gaps in BRAFi-treated DMEC monolayers (Fig 5B). Pre-treatment with high doses of vemurafenib weakened the endothelial barrier against invading tumor cells, resulting in a significantly larger gap area compared with vehicle control treatment (Fig 5C and D). None of the other inhibitors affected the spheroid-induced gap area.

### BRAFi affect vascular junctions in patients

We further aimed to investigate the effects of clinically used BRAFi on the vasculature in skin biopsies from advanced melanoma patients who had been treated with BRAFi. We obtained archived skin biopsies before and during therapy from one patient who had received

**Figure 2. BRAFi disrupt the endothelial phosphoproteome.**
Mass spectrometry-based phosphoproteomics data of dermal microvascular endothelial cells treated with vehicle control (DMSO), 10 $\mu$M of vemurafenib (V10), dabrafenib (D10), encorafenib (E10), PLX8394 (P10), or 100 $\mu$M vemurafenib (V100) for 1 h. **(A)** Z-scored phosphosite abundance per condition (n = 3 donors). **(B)** Overlaps of significantly up- (red) or downregulated (blue) phosphoproteins among treatments relative to the DMSO control (Limma, logFC ± 1, $P \le 0.05$). **(C)** Phosphoprotein abundance of treatments compared with the DMSO control. **(D)** Reactome pathway enrichment analysis of proteins with a significantly altered phosphorylation status, listed according to $P$-value and enrichment ratio calculated as entities found/total number of entities in the pathway.

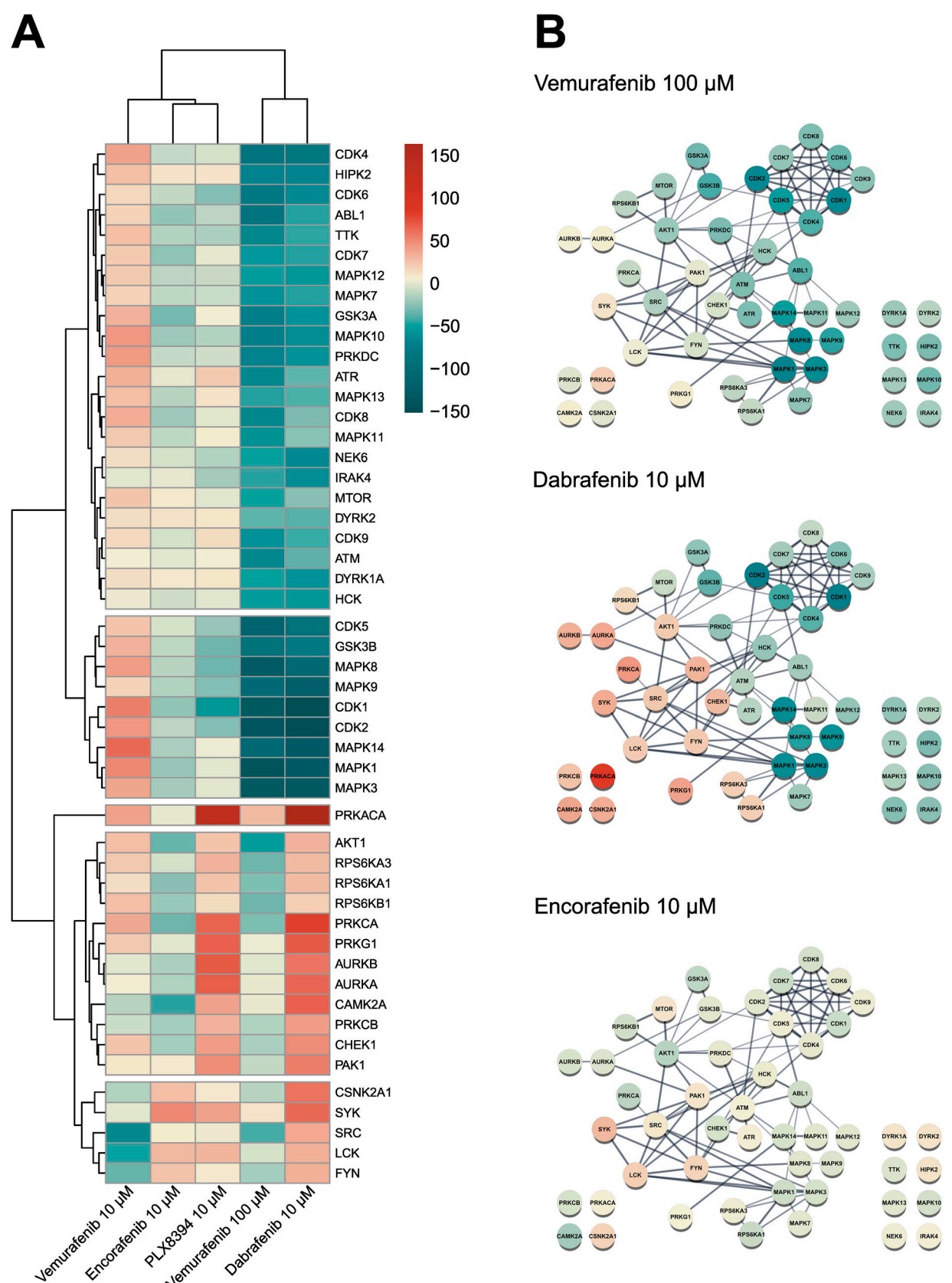

**Table 1. Comparison of KinSwing activity predictions with previously published data on directly bound off-targets of BRAFi (Klaeger et al, 2017).**

| Kinase | V100 | V10 | D10 | E10 |
|---|---|---|---|---|
| BRAF | −29.36 | 14.43 | −9.91 | 6.98 |
| TGFBR2 | −11.77 | 0.00 | 2.28 | – |
| PTK6 | −14.31 | −13.47 | 6.86 | – |
| CDK4 | – | – | −81.97 | −8.65 |
| IRAK1 | – | – | −22.51 | 1.26 |
| CDK1 | – | – | −151.71 | – |
| CDK2 | – | – | −149.28 | – |
| CDK5 | – | – | −93.04 | – |
| CDK6 | – | – | −63.19 | – |
| HCK | – | – | −53.92 | – |
| ABL1 | – | – | −46.74 | – |
| JAK2 | – | – | −34.73 | – |
| ABL2 | – | – | −27.37 | – |
| YES1 | – | – | −26.47 | – |
| MET | – | – | −14.02 | – |
| MELK | – | – | −5.41 | – |
| TGFBR1 | – | – | −1.60 | – |
| ULK1 | – | – | 0.93 | – |
| LYN | – | – | 3.71 | – |
| NEK1 | – | – | 4.17 | – |
| PRKD2 | – | – | 17.05 | – |
| CAMK4 | – | – | 26.79 | – |
| LCK | – | – | 32.89 | – |
| FYN | – | – | 33.49 | – |
| CSNK1A1 | – | – | – | −3.29 |
| GSK3A | – | – | – | −31.07 |
| GSK3B | – | – | – | −10.07 |
| MAPK8 | – | – | – | −12.20 |
| MAPK9 | – | – | – | −13.79 |
| MAPKAPK2 | – | – | – | −13.00 |
| NLK | – | – | – | 0.00 |

Predicted activity scores scaled according to Fig 3A are shown for kinases that were identified in both datasets. Positive values denote increased kinase activity, whereas negative values denote decreased activity, and a value of 0 denotes presence in the Klaeger dataset but no predicted change in kinase activity compared with the DMSO control. Empty cells indicate that the respective kinase was not directly bound in the Klaeger dataset.

vemurafenib monotherapy, one with vemurafenib + cobimetinib, and three patients who had been treated with dabrafenib + trametinib. Tissue sections were then subjected to immunofluorescence staining for vascular markers and quantification of signal intensity was performed in tumor-adjacent skin. Our image analysis showed that within the VE-cadherin-positive endothelium, the signal of tight junction protein claudin-5 was decreased upon vemurafenib monotherapy (72% during treatment versus before, patient #1), whereas the combinations of vemurafenib + cobimetinib and dabrafenib + trametinib did not have strong effects (Table 2). In podoplanin-positive lymphatic vessel walls, we found a decrease in both VE-cadherin and claudin-5 signal upon vemurafenib monotherapy (62% and 41% during treatment versus before, respectively). Representative images of arteries, capillaries, and lymphatic vessels in patient #1 showed the loss of VE-cadherin and claudin-5 signal intensity during vemurafenib treatment (Fig 6A). Sample autofluorescence images from the same positions show no overlap between the signal from junctions and tissue autofluorescence (Fig 6B). Patients who received either of the combination treatments did not show the same effects (Table 2 and Fig S8). Of note, patient #5 was matched with a control sample from a different patient and generally displayed higher fluorescence intensity values during treatment for all markers and quantification masks.

# Discussion

Targeted therapies aimed at mutant BRAF and downstream MAPK signaling components are effective treatments against BRAF-V600E/K-positive melanoma. Currently, three different inhibitors against advanced BRAF-mutant melanoma are clinically approved for therapy: vemurafenib, dabrafenib, and encorafenib. They have different pharmacodynamic profiles that are connected to their inhibitory potency towards mutant BRAF and off-target effects. A detailed review from 2019 discusses the differences among clinically used BRAF and MEK inhibitors, regarding their pharmacodynamics and particularly their adverse event profiles (Heinzerling et al, 2019). Vascular endothelial cells come in direct contact with high inhibitor concentrations and could influence treatment outcomes; however, studies that investigate the effects of BRAFi on the vascular endothelium are lacking. Therefore, our findings of differential effects of clinically used BRAFi on endothelial cells inform the field about potential pathways that could elicit side effects or provide a protumorigenic microenvironment.

## Paradoxical MAPK activation

We observed that inhibitors approved for clinical use increased ERK phosphorylation in DMEC in a concentration-dependent manner, which corresponds to clinical dosing: The $C_{max}$ of vemurafenib (61.4 µg/ml ≙ 125.32 µM) is ~40 times higher than that of dabrafenib and 10 times higher than encorafenib (European Medicines Agency, 2012, 2013, 2018). For endothelial cells, the concentration of BRAFi measured in the patient circulation is critical. The current treatment

**Figure 3. BRAFi differentially affect endothelial kinase signaling.**
**(A)** Predicted kinase activity scores were computed from the phosphoproteomics dataset with KinSwingR. Weighted score for predicted activity of the 50 most differentially regulated kinases across all treatments compared with vehicle control. Scale ≙ Swing score. **(B)** STRING physical subnetwork visualization of the same 50 kinases, Scale ≙ Swing score.

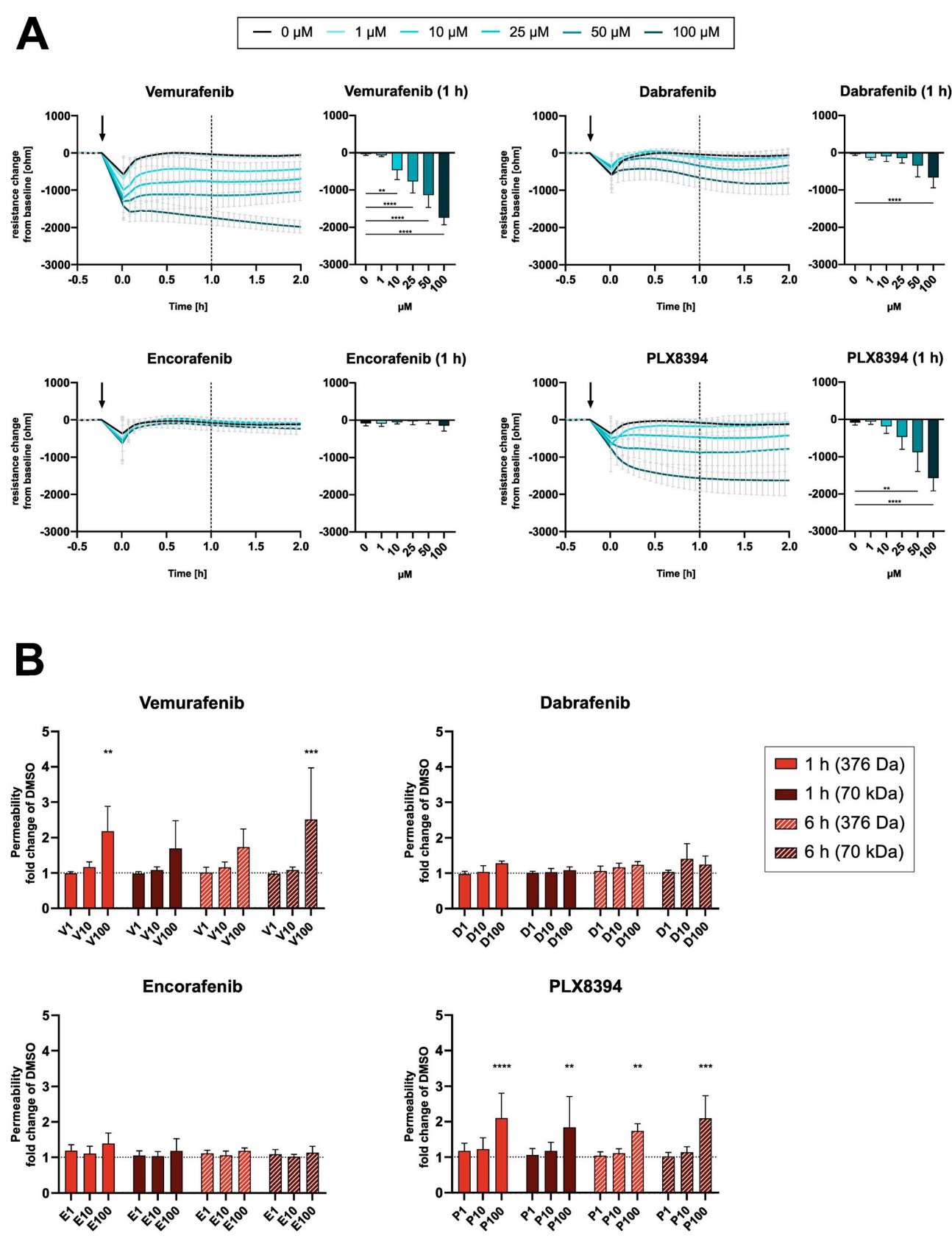

regimen for BRAFi is not adjusted according to the weight or biological sex of the patient (Garbe et al, 2022). This has been reported as a potential factor for increased AEs and dose modifications, especially in women or patients with low body weight (Hopkins et al, 2020). However, these groups also experienced a benefit from a higher exposure to BRAFi and have shown higher overall survival (Vellano et al, 2022). In our study, we could show that clinically relevant BRAFi concentrations represent a sensitive balance between paradoxical activation and inhibition of the MAPK pathway in endothelial cells: In cell culture, concentrations of 10 $\mu$M vemurafenib were necessary to induce paradoxical ERK activation. In comparison, dabrafenib and encorafenib induced ERK phosphorylation already at a lower dose of 1 $\mu$M. This paradoxical MAPK activation did not occur in DMEC treated with the so-called "paradox breaker" PLX8394, a next-generation BRAFi designed to specifically interfere with the dimerization dynamics of mutant BRAF (Basile et al, 2014). The phenomenon of paradoxical MAPK activation has been extensively investigated in BRAF WT cancer cells, especially in the presence of upstream NRAS mutations (Hatzivassiliou et al, 2010; Oh et al, 2016). Over years, there have been increased efforts to elucidate the response to BRAFi not only in tumor cells but also in other cells in or outside of the TME. For example, BRAFi-induced paradoxical ERK activation has been shown in fibroblasts, keratinocytes, and immune cells (Callahan et al, 2014; Escuin-Ordinas et al, 2016; Corrales et al, 2021). This has cell-type specific functional consequences and could play a crucial part in the outcome of BRAFi treatment. A previous publication suggested that the effects of BRAFi on the TME could be contributing to melanoma clearance by improving T-cell infiltration (Wilmott et al, 2012). However, another study showed that paradoxical MAPK signaling in macrophages of the TME could also have tumor-protective effects by promoting resistance mechanisms (Wang et al, 2015).

### Off-targets in endothelial cells

Apart from effects within their target pathway, kinase inhibitors can typically also bind and interfere with other kinases. For example, vemurafenib is known to target not only mutant BRAF but also CRAF, SRMS, and ACK1 with a similar $IC_{50}$ (18–48 nM) and numerous other kinases in the low $\mu$M range in cell-free assays (Bollag et al, 2010). The plasma levels of vemurafenib in patients are by far exceeding thresholds for interfering with a broad range of kinases. This suggests that, apart from MAPK, other signaling pathways would also be affected by BRAFi treatment. To gain deeper insights into kinase signaling dynamics of BRAFi treatment, we performed MS-based proteomics and phosphoproteomics of DMEC treated with the respective inhibitors. We observed no changes in protein abundance, but all used BRAFi had considerable effects on phosphorylation after 1 h of treatment. To our surprise, each BRAFi affected a specific subset of phosphoproteins

inside and outside of the MAPK pathway. Vemurafenib in the lower concentration (10 $\mu$M) caused only minor alterations in the phosphoproteome of endothelial cells, but the higher dose of 100 $\mu$M had a similarly strong effect as 10 $\mu$M of the other BRAFi, which also underlines the pharmacodynamic differences among these inhibitors. The different effects of BRAFi suggest that phosphosites are altered by off-target kinases outside of the MAPK pathway. This hints at a considerable amount of polypharmacology, or target promiscuity, which describes the capacity of an inhibitor to bind more than one target. Target promiscuity can be attributed to the fact that most kinase inhibitors attack the ATP-binding pocket of their target, which is structurally similar among kinases and other enzymes (Tong & Seeliger, 2015; Karoulia et al, 2017). This phenomenon can have detrimental but also beneficial aspects, especially in drug repurposing, but also in complex diseases such as cancer, where concomitant manipulation of oncogenic pathways could either impede or improve treatment efficacy (Kabir & Muth, 2022). For example, recent publications have identified mTOR signaling and the SEMA6A/RHOA/YAP axis as off-target mechanisms in BRAFi-associated tumor-protective effects of fibroblasts in the TME (Seip et al, 2016; Loria et al, 2022). The relevance of effects in the TME is evident and we are first to describe the molecular consequences of second and third-generation BRAFi treatment on the vascular endothelium.

To truly understand the promiscuous nature of therapeutic agents, a comprehensive analysis of on- and off-target effects is necessary, in which proteomics and PTMomics play a central role (Zecha et al, 2023). A study by Klaeger et al investigated the target promiscuity of clinical kinase inhibitors with a competitive affinity assay (kinobeads) paired with MS to assess which proteins would be bound by individual kinase inhibitors in cancer cell lysates (Klaeger et al, 2017). Comparing their datasets with our kinase activity predictions in endothelial cells, we found notable parallels between physical binding and activity regulation for vemurafenib, dabrafenib, and encorafenib. Although physical binding affinity correlated with kinase inhibition in most cases, some of the kinases that were bound by dabrafenib in the Klaeger dataset, including CAMK4, FYN and LCK showed a higher predicted activity in endothelial cells. In addition, BRAF activity was increased after treatment with 10 $\mu$M, but decreased with 100 $\mu$M vemurafenib, highlighting a dynamic dose response that is reflected in the results of Fig 1D and could also apply to off-target kinases. A comparison of these two datasets highlights two aspects of BRAFi dynamics in human cells: predicted kinase activity in an intact layer of live endothelial cells complements physical binding data in cancer cell lysates. It allowed us to identify inhibiting and activating off-targets that overlapped between the datasets. Together with the above cited published studies about off-targets of BRAFi, our findings provide additional support to the hypothesis that effects cannot be attributed to aberrant MAPK signaling alone but also arise from target promiscuity.

**Figure 4. BRAFi disrupt endothelial barrier function.**
**(A)** electrical cell-substrate impedance sensing real-time measurements of electrical barrier resistance in a dermal microvascular endothelial cells monolayer upon BRAFi treatment, displayed as resistance change (ohm) from the time of inhibitor addition (mean ± SD, n = 5–10 biological replicates in four separate experiments). **(B)** Permeability of fluorescently labelled tracers Na-Fluorescein (375 Da) and TRITC-dextrane (70 kD) after 1 and 6 h of BRAFi treatment (n = 3–4 experiments with three biological replicates each). Results are depicted as fold change of the DMSO control (mean ± SD). Significance was tested using two-way ANOVA and Dunnett's test for multiple comparisons with the DMSO control.

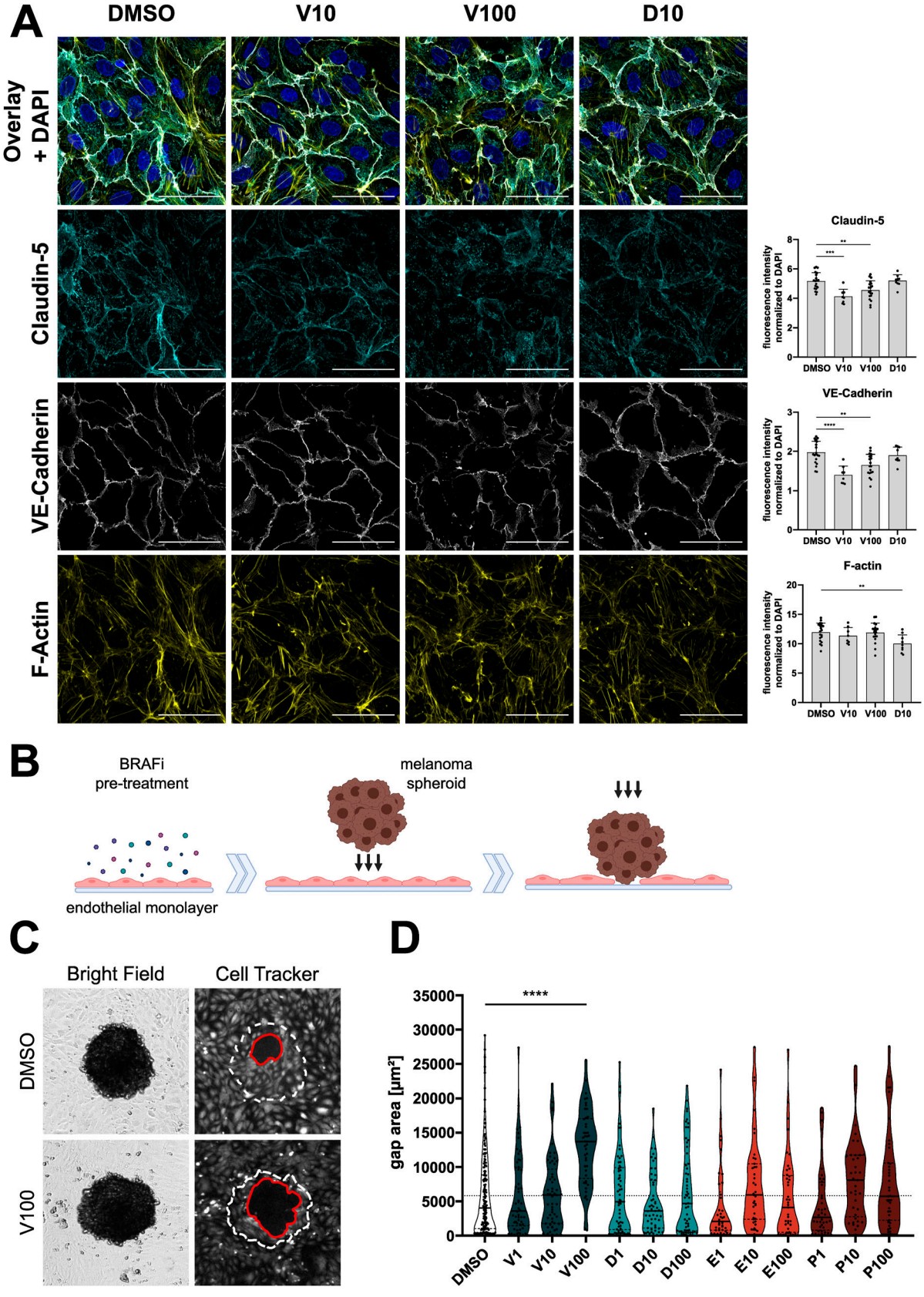

**Table 2.  Vascular junction marker quantifications in tumor-adjacent skin of melanoma biopsies before and during treatment with indicated inhibitors.**

| Patient # | 1 | 2 | 3 | 4 | 5 |
|---|---|---|---|---|---|
| Treatment | Vemurafenib monotherapy | Vemurafenib + cobimetinib | Dabrafenib + trametinib | Dabrafenib + trametinib | Dabrafenib + trametinib |
| Within VE-cadherin⁺ vessels | | | | | |
| Claudin-5 (%) | 72 | 93 | 92 | 92 | 180 |
| Within podoplanin⁺ vessels | | | | | |
| VE-cadherin (%) | 62 | 133 | 131 | 118 | 169 |
| Claudin-5 (%) | 41 | 122 | 106 | 98 | 221 |
| Overall fluorescence intensity | | | | | |
| VE-cadherin (%) | 129 | 94 | 98 | 151 | 104 |
| Claudin-5 (%) | 134 | 98 | 86 | 179 | 111 |
| Podoplanin (%) | 121 | 96 | 95 | 121 | 123 |
| Total quantified tissue area | | | | | |
| Before treatment ($\mu m^2$) | 58,019,380 | 62,369,079 | 25,270,836 | 12,909,993 | 12,445,770 |
| During treatment ($\mu m^2$) | 9,740,902 | 30,001,823 | 6,542,780 | 4,372,562 | 11,464,887 |

Fluorescence signal intensities of vascular markers during treatment are displayed as % of the intensity values before treatment within the same patient, or a matched control. Values were quantified within non-tumor areas that were positive for VE-cadherin or podoplanin, to specifically measure intensities within all vessels or lymphatic vessels, respectively. Overall intensity refers to the mean fluorescence within the entire region of interest, including background and the stromal compartment.

The heterogenous alterations of protein targets among different BRAFi were also reflected in our analysis of enriched signaling pathways in BRAFi-treated endothelial cells. We observed that each BRAFi manipulated its individual set of pathways. For example, we observed enriched terms involving RHO GTPase signaling particularly in samples treated with 100 $\mu M$ vemurafenib. The importance of RHO GTPases in endothelial homeostasis, especially in angiogenesis and permeability, has been discovered many years ago (Wojciak-Stothard et al, 1998; Carbajal & Schaeffer, 1999; Van Nieuw Amerongen et al, 2000). It is known that RHOA regulates vascular permeability by interacting with the cytoskeleton at the site of endothelial junctions, which destabilizes cell-cell contacts (van Buul & Timmerman, 2016; Reinhard et al, 2017). Surprisingly, only dabrafenib treatment was associated with enriched terms regarding RAS and RAF signaling. These findings highlight the distinct off-target effects of BRAFi in endothelial cells.

### Functional implications

There is not much literature regarding functional effects of BRAFi on endothelial cells, apart from one recent publication that investigated the effect of 28 clinically used kinase inhibitors on endothelial permeability (Dankwa et al, 2021). However, they did not include any BRAF-specific inhibitors, except the first-generation inhibitor sorafenib, which induced a weakly barrier-disruptive phenotype.

In this study, we observed that vemurafenib and PLX8394 induced a dose-dependent breakdown of electrical barrier resistance as well as a hyperpermeability for low and high molecular weight fluorescent tracers. High doses also interrupted the integrity of endothelial tight and adherens junctions, which were linked to the actin cytoskeleton. The effects of BRAFi on barrier resistance and cell-cell junctions were comparable between endothelial cells of blood and lymphatic origin. Thus, our functional results confirmed what would be expected from our phosphoproteome analysis and the above cited literature, especially for altered RHO GTPase signaling. However, we did not observe a correlation between paradoxical ERK activation and functional response. Dabrafenib and encorafenib clearly induced a paradoxical ERK activation, but only vemurafenib treatment caused severe endothelial barrier dysfunction. In addition, the dimerization inhibitor PLX8394 did not induce paradoxical MAPK signaling but had similar effects on endothelial function

**Figure 5.  BRAFi differentially affect endothelial junctions and resilience against tumor cell invasion.**
**(A)** Immunofluorescence images of confluent dermal microvascular endothelial cells (DMEC), treated with DMSO, vemurafenib, or dabrafenib for 1 h. Cyan = claudin-5, white = VE-Cadherin, yellow = F-actin. Scale bars = 50 $\mu m$. Right panel: Quantification of overall fluorescence intensities of junctional markers, normalized to the DAPI signal intensity, depicted as mean + SD (n = 9–20 images per condition). **(B)** Fluorescently labelled DMEC monolayers were treated with BRAFi for 6 h, before incubation with melanoma spheroids for 6 h. Melanoma cells breached the endothelial barrier and lead to gaps in the monolayers. **(C)** Brightfield and fluorescence images of melanoma spheroids on top of DMEC monolayers. Gaps (red line) were measured in the endothelial monolayer beneath spheroids (white dotted line). **(D)** The area of gaps is depicted as mean ± SD (n[treatment] = 37–60 spheroids, n[control] = 176 spheroids, from at least three independent experiments). For statistical analysis, all treatments were compared with the DMSO control.

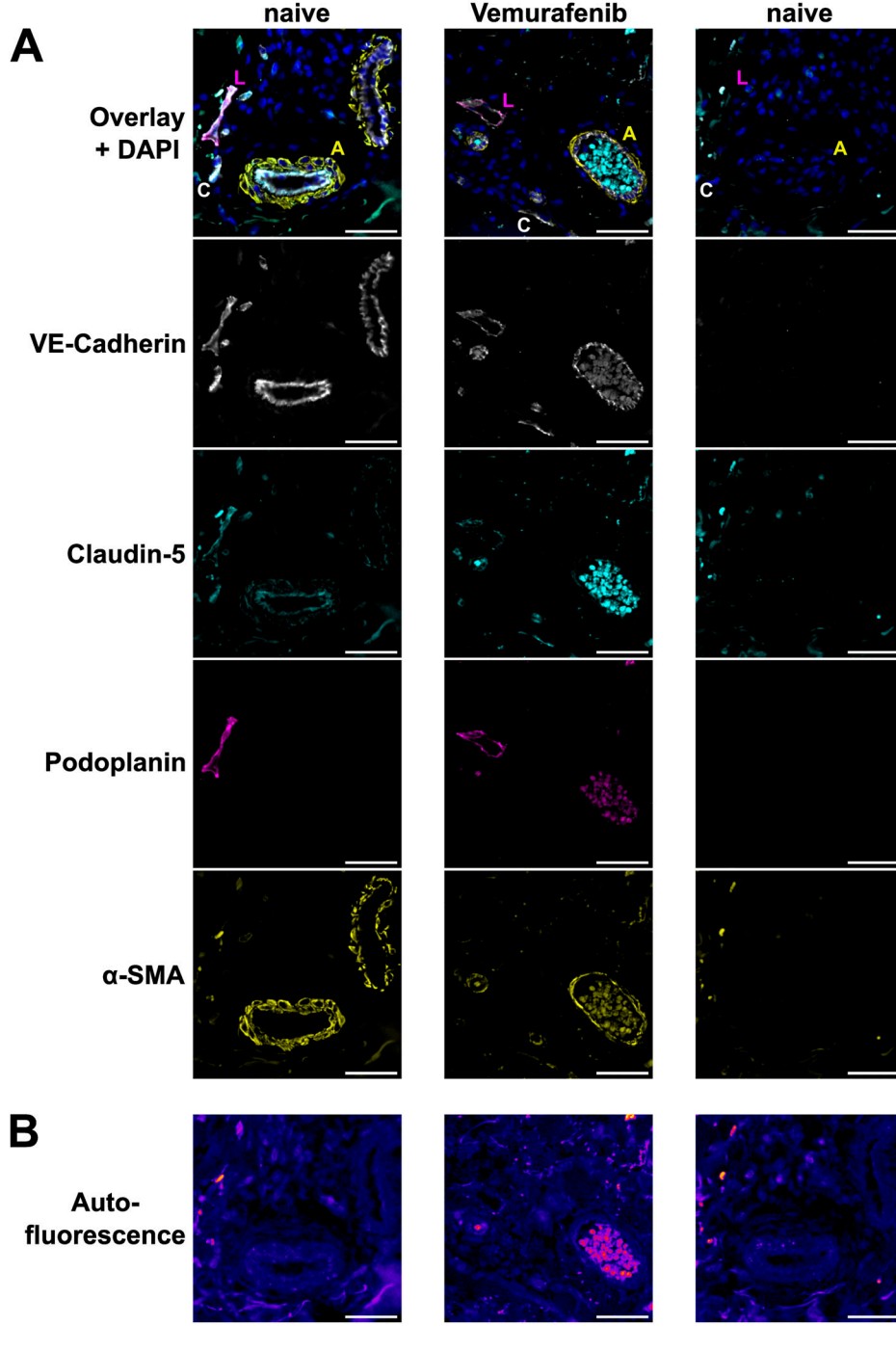

**Figure 6. Effect of vemurafenib on patient vessels.**
**(A)** Immunofluorescence images of vascular markers in skin biopsy sections of patient #1 before (naive) and during vemurafenib monotherapy. Markers are VE-cadherin (white), claudin-5 (cyan), podoplanin (magenta), and α-smooth muscle actin (yellow). Letters specify vessel types in the overlay images as follows: artery (A), capillary (C), lymphatic (L). Scale bars = 50 μm. **(B)** Sample Autofluorescence of the same tissue areas as depicted in (A). Scale bars = 50 μm.

as vemurafenib. Despite the widespread assumption that paradoxical MAPK signaling is mainly responsible for unwanted side effects from data in stromal cells (Adelmann et al, 2016), we provide proof that this is not the case for vascular endothelial cells.

In addition, we investigated the barrier resistance of endothelial cells against tumor cell spheroids based on a previous model of tumor cell invasiveness (Holzner et al, 2016). In our experimental setup, only a high dose of vemurafenib significantly weakened the endothelium against melanoma cell spheroids. Although this simplified model does not account for many factors involved in metastasis formation in vivo, it gives insights about vemurafenib in facilitating the extravasation of tumor cells. Indeed, it has been previously shown that vemurafenib treatment was associated with a higher metastatic burden in a drug-resistant melanoma mouse model (Obenauf et al, 2015) but long-term data from clinical studies on this aspect in human patients are missing.

Notably, dabrafenib and encorafenib had no or minimal effects on the endothelial barrier function even at high concentrations, despite inducing paradoxical MAPK activation and affecting multiple off-target pathways. Comparing the functional results with our phosphoproteomics data, we are not able to pinpoint exactly which off-targets are responsible for adverse effects on endothelial barrier function yet, but we propose closer investigation of promising candidates such as RHOA, CDKs, PKA, or GSK3-$\beta$ and interactions between them based on the data in this study.

### Clinical consequences

The AEs documented for clinically approved BRAFi are distinct from one another, especially when combined with their corresponding MEKi. A number of cutaneous and gastrointestinal AEs are common among all inhibitors, whereas other events occur more frequently with particular BRAFi or MEKi. For example, QT prolongation was observed in up to 7% of patients undergoing vemurafenib mono-therapy (Flaherty et al, 2014; Heinzerling et al, 2019), whereas other cardiovascular events such as pulmonary embolism, arterial hypertension, and decreased left ventricular ejection-fraction have been linked to MEKi (Abdel-Rahman et al, 2016; Mincu et al, 2019). Vasculitis has been described sporadically as an AE in vemurafenib-treated patients (Heinzerling et al, 2019).

We aimed to translate our findings from cell culture into a clinical context by investigating junctional markers of dermal vessels of pre- and on-treatment biopsies from cutaneous metastases of melanoma patients who had received BRAFi therapy. Our inclusion and exclusion criteria yielded five eligible patients who were treated at our clinic either with vemurafenib mono-therapy (one patient), vemurafenib + cobimetinib (one patient), or dabrafenib + trametinib (three patients). Immunofluorescence showed a decrease in endothelial junction markers (VE-cadherin, claudin-5) upon vemurafenib monotherapy, whereas the combination treatments did not have the same effect. These results demonstrate vessel damage upon vemurafenib therapy in one patient, which is coherent with our findings in cultured human endothelial cells. Thus, detrimental effects on endothelial cells could be a potential explanation for the higher AE rate in vemurafenib compared with other BRAFi-treated patients. However, further studies with larger sample sizes are needed to specifically characterize vascular-specific effects of new BRAFi and their consequences for AEs in patients. It is important to note that vemurafenib is barely prescribed to patients nowadays because dabrafenib and encorafenib, together with their respective MEKi, exhibit superior response rates and toxicity profiles and show a reduced occurrence of secondary neoplasms, when compared with vemurafenib (Heinzerling et al, 2019; Garbe et al, 2022). Current clinical guidelines recommend targeted therapies as a second-line treatment after immune checkpoint inhibitor (ICI) therapy for advanced melanoma, although the optimal sequencing strategy for different patient groups is still investigated (Keilholz et al, 2020). Interestingly, the next-generation dimerization inhibitor PLX8394 showed very promising results in preclinical studies, but no clinical data (Phase I/IIa trial NCT02012231, Phase I/IIa trial NCT02428712) are available to date.

In conclusion, we present evidence that inhibitors against mutant BRAF have considerable effects on the vascular endothelium.

Although clinically approved BRAFi induced paradoxical MAPK activation in endothelial cells, their off-target spectra are diverse. This is also reflected in their functional impact on the endothelium. Especially vemurafenib substantially disrupted endothelial barrier function. Therefore, together with the off-target profiles acquired by phosphoproteomics, our results provide proof that BRAFi disrupt endothelial homeostasis. This could give insights into the mechanisms that are responsible for AEs. Future therapeutic developments and clinical studies should consider the target promiscuity of kinase inhibitors in the TME, including the vasculature. Better knowledge of the response to BRAFi in tumor cells and cells of the TME seems critical for future developments and could help to find even better treatment options for specific patient groups.

# Materials and Methods

### Antibodies and reagents

The BRAFi used in this study were vemurafenib (S1267), dabrafenib (S2807), encorafenib (S7108), and PLX8394 (S7965), all purchased from Selleckchem. DMSO (D2650; Sigma-Aldrich) at a concentration of 0.1% was applied as a vehicle control. All antibodies used for this study are presented in Table 3.

### Cell culture

Human DMEC were isolated from freshly discarded foreskin of pediatric patients undergoing circumcisions at the Department of Pediatrics of the Medical University of Vienna. The protocol was approved by the Ethics Committee of the Medical University of Vienna (1621/2020) in accordance with the Declaration of Helsinki. Written informed consent was obtained from the patients' legal guardians. The collected tissues were cut into thin strips before incubation with Dispase (CLS354235; Merck) for 20 min at 37°C. After gentle elimination of the epidermal layer, cells were dislodged with a cell scraper in Endothelial Cell Growth Medium MV (EGM-MV; C-22020; Promocell) with the respective supplement mix, 15% FCS (10500-064; Gibco), and 50 $\mu$g/ml Gentamicin (15710-049; Gibco). The cell suspension was centrifuged at 123$g$ for 10 min, pelleted cells were resuspended in fresh medium and seeded onto a six-well plate. Until the first passaging, 100 $\mu$g/ml primocin (ant-pm-1; InvivoGen) was added to the culture medium. Before the first passaging, cells were sorted with Dynabeads for CD31 (11155D; Invitrogen) to enrich endothelial cells and eliminate contaminating fibroblasts. DMEC were routinely cultured in EGM-MV at 37°C, 5% $CO_2$ and used for experiments between passage 3 and 8.

For experiments specifically comparing endothelial cells of blood and lymphatic origin, DMEC from three donors were sorted as follows: Confluent DMEC were washed once with PBS and then detached using trypsin. Cell suspensions were washed once with PBS and then resuspended in 1% BSA in PBS. Cells were stained with directly labelled antibodies for CD31 and podoplanin for 30 min with gentle agitation at 4°C. The suspension was washed once with PBS and was then resuspended in PBS for sorting on a FACSAria Fusion Flow Cytometer (BD Biosciences). Viable single cells were identified either as BEC (CD31-positive, podoplanin-negative) or LEC

**Table 3. Primary and secondary antibodies used in this study.**

| Target | Supplier | Cat. Nr. | Dilution |
|---|---|---|---|
| p-ERK1/2 (T202/Y204) | CST | 9101 | 1:1,000 (WB) |
| ERK1/2 | CST | 4695 | 1:1,000 (WB) |
| GAPDH | Abcam | ab181602 | 1:20,000 (WB) |
| Rabbit IgG (IRDye® 800CW-conjugated second step) | LI-COR | 925-32213 | 1:15,000 (WB) |
| Rabbit IgG (IRDye® 680RD-conjugated second step) | LI-COR | 925-68073 | 1:15,000 (WB) |
| VE-cadherin | Beckman Coulter | IM1597 | 1:200 (IF cells) |
| VE-cadherin | CST | 93467 | 1:200 (IF tissue) |
| Claudin-5 (AF488-conjugated) | Invitrogen | 352588 | 1:200 (IF cells) |
| Claudin-5 | Invitrogen | 352500 | 1:200 (IF tissue) |
| α-SMA (FITC-conjugated) | Sigma-Aldrich | F-3777 | 1:5,000 (IF tissue) |
| Podoplanin (AF647-conjugated) | BioLegend | 337008 | 1:500 (IF tissue), 1:200 (FC) |
| Prox1 | R&D Systems | AF2727 | 1:200 (IF cells) |
| CD31 (PE-conjugated) | eBioscience | 12-0319-42 | 1:200 (FC) |
| ICAM-1 (PE-conjugated) | BD Biosciences | 347970 | 1:100 (FC) |
| E-Selectin (FITC-conjugated) | Fitzgerald | 61R-CD62ebHUFT | 1:50 (FC) |
| Mouse IgG (AF546-conjugated second step) | Life technologies | A-11030 | 1:500 (IF cells/tissue) |
| Rabbit IgG (AF594-conjugated second step) | Life technologies | A-32754 | 1:500 (IF tissue) |
| Goat IgG (AF647-conjugated second step) | JIR | 705-605-147 | 1:400 (IF cells) |

Primary and secondary antibodies used in the present study were purchased from the following suppliers: Cell Signaling Technologies (CST), Abcam, LI-COR, Beckman Coulter, Invitrogen, BioLegend, R&D Systems, Life technologies, or Jackson Immunoresearch (JIR). Antibody dilutions are indicated per method as WB for Western blot, FC for flow cytometry and IF for immunofluorescence. CST stands for Cell Signaling Technologies and JIR for Jackson Immunoresearch.

(double-positive) and sorted in separate tubes containing EGM-MV medium. Sorted cells were washed once with fresh medium.

Patient-derived melanoma cell lines (Pirker et al, 2003; Puujalka et al, 2016), including VM15 (NRAS mutant), VM53 (BRAF and NRAS WT), VM21, and VM48 (both BRAF mutant), were cultured in RPMI-1640 (21875-034) supplemented with 10% FCS and 50 U/ml streptomycin-penicillin (15070-063), which were all from Gibco. All cells were maintained in a humidified atmosphere containing 5% $CO_2$ at 37°C and passaged at 90% confluence.

## Western blot

After treatment with the indicated BRAFi for 1 h, DMEC or melanoma cells were washed with ice-cold PBS and lysed on ice using a radioimmunoprecipitation buffer (RIPA, containing 50 mM Tris–HCl, 150 mM NaCl, 1% NP-40, 0.5% sodium deoxycholate, 0.1% SDS, and 1 mM EDTA) supplemented with protease inhibitor (P8340; Sigma-Aldrich) and phosphatase inhibitor cocktails (P5726, P0044; Sigma-Aldrich). Lysates were centrifuged at 18,000$g$ for 15 min at 4°C and supernatants were used for further analysis. Protein concentrations were determined using Bradford Protein assay (500-0006; Bio-Rad) according to the manufacturer's protocol. Per sample, 20 $\mu$g of protein was mixed with reducing Laemmli buffer and denatured for 5 min at 95°C. After SDS–PAGE, proteins were transferred onto a nitrocellulose membrane (GE Healthcare Life Sciences) by wet blotting. Equal protein loading was confirmed by staining with Ponceau-S (33427; Serva). After blocking with 5% BSA (A2153; Sigma-Aldrich) in TBS buffer containing 0.1% Tween-20 (TBST), membranes

were washed with TBST and incubated overnight at 4°C with the indicated primary antibodies diluted in TBST and 5% BSA. After washing with TBST, the membranes were incubated with fluorescent secondary antibodies (LI-COR), diluted in 5% milk powder (70166; Sigma-Aldrich) in TBST for 1 h at RT. Blots were imaged and analyzed using Odyssey CLx and Image Studio (version 5.2) from LI-COR.

## Phosphoproteomics

### Sample preparation and TMT labeling

Confluent DMEC from three donors were treated with the indicated inhibitor concentrations or vehicle control for 1 h. After washing with ice-cold PBS, cells were lysed on ice with 1% SDS and protease/phosphatase inhibitor cocktails. Lysates were homogenized by sonication. Lysates were incubated with Dithiothreitol (DTT) for 30 min at 37°C and alkylated with 20 mM iodoacetamide for another 30 min. Proteins were purified by precipitation with ethanol-acetone, and protein pellets were resolubilized in 1% sodium deoxycholate (SDC) in 50 mM triethylammonium bicarbonate (TEAB) buffer, pH 8. For each sample, 150 $\mu$g proteins were brought to final volume of 95 $\mu$l (in SDC-TEAB buffer) and digested, firstly with Lysyl-endopeptidase (0.01 AU; FUJIFILM Wako Pure Chemicals Corp.) for 2 h at 37°C and subsequently with trypsin (7.5 $\mu$g per sample) for 4 h at 37°C. Peptides in each sample were labeled with one of the tags of TMTpro 18-plex labeling kit (A52045; Thermo Fisher Scientific) according to the manufacturer's instructions. All 18 samples were then pooled in a 1:1 total TMT channel intensity ratio, measured by

high resolution LC-MS2. Pooled samples were acidified (pH < 2) with formic acid to precipitate SDC and the collected supernatant was lyophilized.

### TiO₂ phosphopeptide enrichment

Phosphopeptides were isolated using $TiO_2$ affinity chromatography as previously described (Engholm-Keller & Larsen, 2016). Briefly, dried peptides were dissolved in $TiO_2$ loading buffer (80% acetonitrile [ACN], 5% trifluoroacetic acid [TFA] and 1 M glycolic acid, all from Sigma-Aldrich) and incubated with 0.6 mg $TiO_2$ beads (Titansphere, GL Sciences) per 100 µg peptide solution for 30 min at RT on a shaker. The beads were spun down and supernatant was transferred to a new tube with 0.3 mg beads per 100 µg peptide. After 15 min shaking at RT, beads were spun down again, and supernatant was collected in a separate tube. $TiO_2$ beads from both incubations were subsequently washed with 80% ACN/1% TFA and 10% ACN/0.1% TFA. The unbound peptides in the supernatants from incubation and wash steps were combined and stored as "unmodified peptides" (further details below). The $TiO_2$ beads with bound phosphopeptides were resuspended in 100 µl of 100 mM TEAB, pH 8.5 and incubated with PNGase F (1,000 U; New England BioLabs) and Sialidase A (5 mU; Prozyme/Agilent) overnight at 37°C to deglycosylate peptides (Larsen et al, 2007). The phosphorylated peptides were eluted from the $TiO_2$ beads by incubation with 1.5% ammonium hydroxide solution, pH 11.3, for 10 min at RT with vigorous shaking. The beads were spun down and the supernatant was passed through a C8 membrane (3 M Empore; Sigma-Aldrich), to remove any residual beads. The membrane was then washed with 100 µl of 50% ACN to obtain any retained peptide, before the samples were dried. To reduce sample complexity, high-pH reversed-phase fractionation was applied (Boll et al, 2020). Phosphopeptides were dissolved in 20 mM ammonium formate, pH 9.3, and loaded on an Acquity UPLC-Class CSHTM C18 column (Waters). Fractionation was performed on a Dionex Ultimate 3000 HPLC system (Thermo Fisher Scientific), and a total number of 20 concatenated fractions was collected.

### Nano-flow liquid chromatography-mass spectrometry (nLC-MS/MS) analysis

The analysis was performed on an Easy-nLC System (Thermo Fisher Scientific) using buffer A (0.1% formic acid, FA) and buffer B (95% ACN, 0.1% FA) and an Orbitrap Eclipse Tribrid MS (Thermo Fisher Scientific). All fractions were redissolved in buffer A (0.1% FA) and loaded into the in-house made fused silica capillary column setup (18 cm pulled emitter analytical column with 75 µm inner diameter, packed with Reprosil-Pur 120 C18-AQ, 3 µm [Dr. Maisch GmbH]). The peptides were eluted with gradient elution from 2% to 95% buffer B, with a flow rate of 300 nl/min. Ionization was performed by nano-electrospray. Phosphopeptides were analyzed by data-dependent acquisition in positive ion mode mass spectrometry. The m/z scan range for full MS scan was 350–1,500 Da, and intact peptides were detected in the Orbitrap with a resolution of 120,000 full width half maximum (FWHM), a normalized automatic gain control (AGC) target value of 250%, and a maximum injection time of 50 ms. From each full scan, the top 10 most intense precursor ions were selected for higher energy collision dissociation fragmentation with a normalized collision energy (NCE) of 35%. The $MS^2$ was performed with the following parameters: orbitrap resolution of 45,000 FWHM, normalized AGC target

value of 300%, isolation window of 1.2 m/z, dynamic exclusion window of 3 s and a maximum injection time in automatic mode.

### Protein identification and quantification

Protein identification was performed using Proteome Discoverer (version 2.4.0.305; Thermo Fisher Scientific). The search was performed against the UniProtKB/Swissprot database (homo sapiens, release-2022_04/) using an in-house Mascot server (v2.8.2; Matrix Science Ltd) and the built-in Sequest HT search engine. Fixed modifications in the search included TMT-Pro_K (K), TMT-Pro_N term (N-term) and Carbamidomethyl (C), whereas Deamidated (N) and Phosphorylation (S, T, Y) were set as dynamic modifications. Further parameters of the search included a fragment mass tolerance of 0.03 Da, a precursor mass tolerance of 10 ppm and maximum of two missed cleavages. Data filtering was performed using a percolator, with ≤1% false discovery rate (FDR) (peptide and protein level). This resulted in a list of 7,756 master proteins, 11,458 peptide groups (filtered for phosphorylation modification), 12,543 peptide isoforms, 187,269 PSMs and 1,022,760 MS/MS spectra. The abundance values of peptide groups were normalized to the total peptide amount of each channel through Proteome Discoverer. Because the treatments did not cause changes in cellular protein expression but only resulted in minor losses of extracellular matrix components in the 100 µM vemurafenib samples (see Fig S2) phosphopeptide abundances were not further adjusted to respective protein levels.

### Bioinformatic analysis

Differences between treatments and the vehicle control were determined via Limma testing (including paired tests within donors), using the combined statistical testing tool PolySTest (Schwämmle et al, 2020). Phosphosites were considered as significantly altered at a $log^2$-fold change ± 1 and an FDR ≤ 0.05. Reactome (v84; reactome.org) pathway enrichment analysis of significantly altered proteins was performed for each treatment. The resulting lists of pathways were then filtered for hits where at least one treatment fulfilled the thresholds ($P ≤ 0.05$ and strength [entities found/entities in the pathway] ≥ 0.05). Terms involving infectious disease (R-HSA-1169410, R-HSA-9609690, R-HSA-8875360, R-HSA-1169408, R-HSA-8876384) were removed because of contextual inapplicability. Data visualization was performed with R version 4.2.2 (R Foundation for Statistical Computing). In addition, kinase activity prediction was performed using the KinSwingR package, based on phosphopeptide abundance in our dataset and known kinase-substrate interactions from the PhosphoSitePlus database (Engholm-Keller et al, 2019). Swing scores represent an activity index for each kinase, calculated from the abundance of phosphorylated substrates in inhibitor-treated cells, relative to the DMSO-treated control samples. Physical subnetwork visualizations of the 50 most differentially regulated kinases among all treatments were created with the Cytoscape software (version 3.9.1), including the applications Omics Visualizer and STRING, using a 0.6 confidence score cutoff.

### Proteomics of unmodified peptides

After enrichment of phosphopeptides, supernatants from incubation and wash steps were collected and stored as the unmodified

fraction. The unmodified peptides were purified using a commercial HLB cartridge that was equilibrated with ACN and 0.1% TFA, subsequently, before loading and washing the sample with 0.1% TFA. Elution was performed with 65% ACN, 0.1% TFA solution, and samples were dried. Simultaneously to the phosphopeptides, unmodified peptides were also dissolved in buffer A and loaded on an Acquity UPLC-Class CSHTM C18 column (Waters). The fractionation was performed on a Dionex Ultimate 3000 HPLC system (Thermo Fisher Scientific), and 20 concatenated fractions were collected.

Analysis of the unmodified peptides was performed with the same instrumentation as the phosphopeptides. The peptides were eluted with gradient elution using a flowrate of 250 nl/min from 2% to 95% buffer B and were analyzed by data-dependent acquisition and positive ion mode mass spectrometry. The m/z scan range for full MS scan was 350–1,600 Da, and intact peptides were detected in the Orbitrap with a resolution of 120,000 FWHM, an AGC target value of $1 \times 10^6$ ions, and a maximum injection time of 50 ms. Furthermore, the peptides were selected for collision-induced dissociation, using fixed collision energy set to 35%. The fragment ion spectra were recorded using the Ion trap at a resolution of 15,000, an AGC target value of $1 \times 10^5$ MS$^2$ ions and maximum injection time of 150 ms. MS$^3$ spectra were acquired using synchronous precursor selection of 10 precursors. MS$^3$ precursors were fragmented by higher energy collision dissociation with 55% of the collision energy and analyzed using the Orbitrap at 30,000 resolution power with a scan range of 100–500 m/z, an isolation window of 2 m/z, an AGC target value of $1 \times 10^5$ MS$^2$, and maximum injection time of 120 ms with one microscan.

Protein identification and quantification was performed using Proteome Discoverer, as described above. The search resulted in 8,132 master proteins, 49,200 peptide groups, 73,834 peptide isoforms, 176,894 PSMs and 1,523,663 MS/MS spectra. For statistical testing the combined statistical testing tool PolySTest was used and data were visualized with R. Normalized abundances of proteins were analyzed via Limma testing including paired tests within donors. Differences between treatments and the vehicle control were considered statistically significant with log$^2$-fold change ± 1 and an FDR ≤ 0.05.

## ECIS

ECIS (Applied Biophysics) was used to measure barrier resistance of DMEC monolayers. 8W10E+ array plates (72040; ibidi) were coated with 1% gelatin before cell seeding at a density of 15,000 DMEC/cm$^2$. Resistance was measured continuously in a multi-frequency setup. After the resistance at 4,000 Hz reached a stable plateau of >1,000 ohm, endothelial cells were treated and continuously monitored at 250 Hz as previously described (Schossleitner et al, 2016).

## Permeability of fluorescent tracers

DMEC were seeded into transwell inserts (734-2747; VWR International) and cultured until confluence. Indicated treatments were added to the transwell, along with 0.2 μg/ml Na-Fluorescein (376 Da, F6377; Sigma-Aldrich) and 50 μg/ml TRITC-conjugated dextran (70 kD, D1818; invitrogen) tracers. At indicated timepoints,

fluorescence intensity was measured in the medium below the transwell insert with a standard plate reader, using the settings for Fluorescein (excitation: 485 nm, emission: 535 nm) and TRITC (excitation: 540 nm, emission: 600 nm).

## Immunofluorescence

DMEC were seeded on μ-Slide chamber slides (ibidi), grown to 100% confluence, and fixed with 4% PFA for 15 min at RT. After permeabilization in a solution of 20 mM Hepes, 300 mM sucrose, 50 mM NaCl and 0.5% Triton X-100 for 5 min at −20°C, cells were stained with indicated primary antibodies diluted in PBS containing 1% BSA overnight at 4°C and appropriate secondary antibodies for 1 h at RT. Nuclei were stained with DAPI (D9542, 1:1,000; Sigma-Aldrich). Cells were then imaged using a confocal laser scanning microscope (LSM-980; Carl Zeiss) equipped with a Plan-Apochromat 63×/1.40 oil lens. In addition, multiple images per condition were taken in a 10x magnification and the fluorescence intensity of junctional markers was quantified using FIJI software (Fiji is just ImageJ, version 1.54b). Raw fluorescence values were then normalized to the signal of the DAPI channel within the same image.

## Melanoma spheroid-induced gap formation

Based on a previously published in vitro assay of tumor cell invasion (Holzner et al, 2016), BRAF-mutant VM48 melanoma cells (1,500 per well) were seeded in a round-bottom 96-well plate (Greiner Bio-One) in full RPMI containing 0.3% methylcellulose (4,000 cP; M0512; Sigma-Aldrich), followed by centrifugation for 15 min at 335g and 12°C, and incubation for 3–4 d. Meanwhile, DMEC were seeded on μ-Slide four-well chamber slides (ibidi) and grown to 100% confluence. After staining with CellTracker green CMFDA Dye (C2925; Invitrogen), DMEC were pre-treated with the indicated inhibitor concentrations for 6 h before washing with medium. Subsequently, melanoma spheroids were collected, carefully washed and resuspended in EGM2-MV and added onto the endothelial monolayer (~24 spheroids per well). After an incubation of 6 h, chamber slides were scanned with an automated microscope (Cytation 5; Agilent) with a 4x objective and filters for high-contrast brightfield and GFP fluorescence. The area of circular discontinuities within the endothelial monolayer beneath the spheroids was quantified using FIJI software (Fiji is just ImageJ, version 1.54b).

## Histology and immunofluorescence of patient samples

We conducted a comprehensive screening of archived histological samples from melanoma patients who visited the Department of Dermatology at the Vienna General Hospital between 2012 and 2022. The study was conducted according to the principles expressed in the Declaration of Helsinki. We identified a total of 90 patients who met our predefined criteria, as outlined in the ethical protocol approved by the ethics committee of the Medical University of Vienna with approval number 1820/2022. Inclusion criteria were defined as follows: (i) verified diagnosis of melanoma stage IIIA-IVM1d, (ii) BRAF-V600E/K mutation, (iii) therapy with either vemurafenib alone, vemurafenib + cobimentinib, or dabrafenib +

trametinib, and (iv) an age of 18–99 yr at the time of sample collection. Patients were excluded from the study if they received simultaneous treatment with other cancer therapies such as ICIs. Upon further evaluation, only 15 patients had available matching pre- and on-treatment metastatic tissue samples. Of those, we excluded nine patients who did not have skin biopsies available, but metastatic tissue samples from other organs. One sample was not released for research purposes. Consequently, we included a total of five patients, four of which were unique individuals and one was a matched pair based on age, sex, disease stage, and lactate dehydrogenase levels. The patient cohort included three males and two females, with a median age of 70 yr (range: 39–81 yr) and a median lactate dehydrogenase level of 208 U/liter (range: 183–484 U/liter). Their disease stages were classified as IVM1c (n = 4) or IVM1d (n = 1) according to the American Joint Committee on Cancer (AJCC) eighth edition staging system. All included melanomas were tested positive for the BRAF-V600E mutation and had been treated either with vemurafenib monotherapy (n = 1), vemurafenib + cobimetinib (n = 1), or dabrafenib + trametinib (n = 3). From the patients' formalin-fixed paraffin-embedded (FFPE) cutaneous metastatic tissue samples, sections with a thickness of 7 $\mu$m were cut and stained via immunofluorescence for the indicated antibodies (see Table 3), nuclei were counterstained with DAPI. Slides were scanned using a Vectra Polaris imaging system (Akoya Biosciences, Inc.) with a 20x objective. Image analysis and fluorescence intensity quantification was performed in peritumoral tissue areas via the QuPath software (version 0.4.3) (Bankhead et al, 2017).

### Assessment of endothelial cell viability

DMEC were seeded on clear-bottom 96-well plates and grown to 100% confluence. Cells were treated with the indicated inhibitor concentrations for 1 or 6 h, before washing carefully with PBS. DMEC were then stained simultaneously with calcein AM (LIVE) and ethidium homodimer-1 (EthD1, DEAD), both from Thermo Fisher Scientific (LIVE/DEAD Viability/Cytotoxicity Kit for Mammalian Cells, #L3224), as well as NucBlue Live ReadyProbes Reagent (Hoechst 33342, #R37605; Thermo Fisher Scientific) according to the manufacturer's instructions. Staining solutions were decanted and fresh DMEC medium was added to the cells, which were immediately scanned at 37°C with an automated microscope (Cytation 5; Agilent) equipped with a 4x objective and filters for DAPI, GFP, and TRITC fluorescence. The percentage of live and dead cells was then calculated by dividing the number of calcein- and EthD1-positive nuclei by the total number of nuclei via the instrument's software (Gen5, version 3.12).

### Endothelial adhesion markers

DMEC were seeded onto 24-well plates and grown to confluence, before being treated with the indicated inhibitor concentrations or 100 ng/ml bacterial lipopolysaccharide (LPS, L2280; Sigma-Aldrich) for 6 h. After washing cells with PBS, they were detached with trypsin. Cell suspensions were washed with PBS and with 1% BSA.

Each suspension was stained with directly labelled antibodies for ICAM-1 and E-Selectin for 1 h at 4°C. Suspensions were washed with PBS before measurements on a CytoFLEX flow cytometer (Beckman Coulter). Data were analyzed in the CytExpert software (version 2.5).

### Statistical rationale

Unless otherwise specified, differences between treatments and the control were analyzed using one-way ANOVA, corrected with Dunnett's multiple comparisons test in GraphPad Prism (version 8.0.1). Significance levels are depicted in the graphs as follows: $P < 0.05$ (*), $P < 0.01$ (**), $P < 0.001$ (***), $P < 0.0001$ (****).

## Data Availability

The mass spectrometry proteomics and phosphoproteomics data have been deposited to the ProteomeXchange Consortium via the PRIDE partner repository (Perez-Riverol et al, 2022) with the dataset identifiers PXD052251 and PXD052259. Imaging data are available at the BioImage Archive (Sarkans et al, 2018) under the identifiers S-BIAD1169 (immunofluorescence of DMEC) and S-BIAD1170 (immunofluorescence of patient material).

## Supplementary Information

## Acknowledgements

This work has been funded by the Vienna Science and Technology Fund (WWTF) [10.47379/LS18080] to K Schossleitner, by the Comprehensive Cancer Center Vienna—Medical University of Vienna (Cancer Research Initiative Grant to S Bromberger), the European Molecular Biology Organization (Scientific Exchange Grant #9313 to S Bromberger), the City of Vienna (Cancer Research Fund #22077, to K Schossleitner and S Bromberger). Confocal images were obtained at the Core Facility Imaging of the Medical University Vienna. We thank Marion Gröger (†), Sabine Rauscher, Christoph Friedl and Philipp Velicky for their continuous support. Cell sorting was performed at the Core Facility Flow Cytometry of the Medical University Vienna, with the help of Günther Hofbauer. We thank Sofia Greilich for her technical assistance. This study was supported by the Villum Center for Bioanalytical Sciences at the University of Southern Denmark.

### Author Contributions

S Bromberger: conceptualization, data curation, formal analysis, funding acquisition, investigation, visualization, methodology, project administration, and writing—original draft, review, and editing.
Y Zadorozhna: formal analysis, investigation, and writing—review and editing.
JM Ressler: resources, methodology, and writing—review and editing.

S Holzner: methodology.

A Nawrocki: methodology and writing—review and editing.

N Zila: methodology and writing—review and editing.

A Springer: resources.

M Røssel Larsen: resources, supervision, methodology, and writing—review and editing.

K Schossleitner: conceptualization, supervision, funding acquisition, methodology, project administration, and writing—original draft, review, and editing.

## Conflict of Interest Statement

JM Ressler received speaker honoraria from Bristol-Myers Squibb, Roche, Amgen and Novartis and travel support by Sanofi, Roche, and Bristol-Myers Squibb through institution. All other authors declare that they have no conflict of interest.

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
