## [Reviewer comments · Life Science Alliance]

Life Science Alliance

Off-targets of BRAF inhibitors disrupt endothelial signaling and vascular barrier function

Sophie Bromberger, Yuliia Zadorozhna, Julia Ressler, Silvio Holzner, Arkadiusz Nawrocki, Nina Zila, Alexander Springer, Martin Røssel-Larsen, and Klaudia Schossleitner

DOI: <https://doi.org/10.26508/lsa.202402671>

Corresponding author(s): *Klaudia Schossleitner, Medical University of Vienna*

Review Timeline:

Submission Date:	2024-02-22
Editorial Decision:	2024-02-22
Revision Received:	2024-04-26
Editorial Decision:	2024-05-06
Revision Received:	2024-05-17
Accepted:	2024-05-20

Scientific Editor: *Eric Sawey, PhD*

Transaction Report:

Please note that the manuscript was reviewed at *Review Commons* and these reports were taken into account in the decision-making process at *Life Science Alliance*.

Review
COMMONS

Reviews

Ref #1

In general; a major issue that is affecting the whole story is the rather high concentration of Vemurafenib (100 uM) used in the study. The authors do not provide any data describing the viability and function of the endothelial cells after exposure to 100 uM of Vemurafenib. Instead they have chosen two concentrations with a large (10X) difference. Where the cells viable at 100 uM of Vemurafenib? If the endothelial cells were suffering from 100 uM Vemurafenib, they will immediately loose the cell-cell contacts/junctions and thereby any performed permeability assay would be pointless. Furthermore, isolation of skin endothelial cells is at risk to be accompanied with lymphatic endothelial cell contamination. The authors should provide data ensuring that the cells are of >90% endothelial cell purity by checking for PROX1-positive cells together with endothelia cells markers (CD31, VE-cadherin, uptake of AcLDL etc).

The work is of importance in understanding consequences for endothelial cells exposed to BRAF inhibitors used in the clinic using clinically relevant concentrations of the drugs investigated in vitro. If the authors provide with a major revision, the work could be acceptable for publication.

This manuscript addresses an important question; how is the vasculature affected by cancer treatments? It is not unusual that the vascular status is neglected in clinical treatment studies. The manuscript provides valuable phosphoproteomics data of great interest related to this topic. The major weakness of the work is the lack of data verifying the chosen concentrations for the BRAF inhibitors used in the study. There is a great risk that several results based on the 100 uM Vemurafenib treatment (of high impact for the story) are based on cell toxicity due to a high concentration treatment in vitro. Also, the link between the strategy of performed in vitro experiments isn't clear and there is a lack of connecting the in vitro data to the validation performed on melanoma patient tissue biopsies. It is a great strategy to investigate skin biopsies before and after treatment. The precious biopsy material should be more carefully investigated and evaluated.

Audience: after improvement of the manuscript by better presentation of existing data and by additional experiments the work presented would be of interest to a pre-clinical and clinical audience investigating cancer treatments.

Ref #2:

The authors provide a solid story outlining the pitfalls in BRAFi therapy research and the consequences on endothelial vasculature in the treatment of BRAF mutant melanoma. The manuscript details clinical relevance of the research, functional impact to the field, and a thorough discussion on the scope of this work and where it may be lacking, which allows for the opportunity for future directions.

Whether in the context of treating melanoma or any other disease. this manuscript serves as a helpful reminder to pre-clinical and clinical researchers alike in how important it is to factor in the patient as a whole, not just the disease when identifying effective treatment options.

February 22, 2024

Re: Life Science Alliance manuscript #LSA-2024-02671-T

Dr. Klaudia Schossleitner
Medical University of Vienna
Austria

Dear Dr. Schossleitner,

Thank you for submitting your manuscript entitled "Off-targets of BRAF inhibitors disrupt endothelial signaling and differentially affect vascular barrier function" to Life Science Alliance. We invite you to re-submit the manuscript, revised according to your Revision Plan.

Thank you for this interesting contribution to Life Science Alliance. We are looking forward to receiving your revised manuscript.

Sincerely,

B. MANUSCRIPT ORGANIZATION AND FORMATTING:

Response to the reviewers

Manuscript number: LSA-2024-02671-T

Title: Off-targets of BRAF inhibitors disrupt endothelial signaling and vascular barrier function

Authors:

Sophie Bromberger¹, Yuliia Zadorozhna¹, Julia Maria Ressler¹, Silvio Holzner¹,
Arkadiusz Nawrocki², Nina Zila^{1,3}, Alexander Springer⁴, Martin Røssel Larsen²,
Klaudia Schossleitner^{1*}

1 Department of Dermatology, Medical University of Vienna, Vienna, Austria

2 Department of Biochemistry and Molecular Biology, University of Southern Denmark, Odense, Denmark

3 University of Applied Sciences FH Campus Wien, Division of Biomedical Science, Vienna, Austria

4 Department of Pediatric Surgery, Medical University of Vienna, Vienna, Austria

* Corresponding Author

Reviewer #1 (Evidence, reproducibility and clarity (Required)):

Summary:

The author Bromberger and colleagues have submitted a MS # RC-2023-02152 entitled "Off-targets of BRAF inhibitors disrupt endothelial signaling and differentially affect vascular barrier function" for review via Review Commons. In the MS they have investigated four BRAF inhibitors with different pharmacodynamics; Vemurafenib, Dabrafenib, Encorafenib and PLX8394 and their specific effect on vascular endothelial cells but also on melanoma cells. The study is composed of in vitro studies using in-house isolated human dermal endothelial cells. Also, melanoma cells and skin biopsies from 5 melanoma patients were analysed. The authors conclude that the BRAF inhibitor Vemurafenib caused strong effect on the endothelial cells' barrier function in comparison to the other three BRAF inhibitors.

Major comments: major issues affecting the conclusions:

In general; a major issue that is affecting the whole story is the rather high concentration of Vemurafenib (100 μ M) used in the study. The authors do not provide any data describing the viability and function of the endothelial cells after exposure to 100 μ M of Vemurafenib. Instead they have chosen two concentrations with a large (10X) difference. Where the cells viable at 100 μ M of Vemurafenib? If the endothelial cells were suffering from 100 μ M Vemurafenib, they will immediately loose the cell-cell contacts/junctions and thereby any performed permeability assay would be pointless.

We thank the reviewer for this valuable comment. Indeed, we did not show viability data upon BRAFi treatment in the original manuscript and we have corrected this in our revised version. As can be seen in the new Supplementary Figure S3, we performed LIVE/DEAD assays of endothelial cells treated with multiple concentrations of BRAFi for the relevant timepoints. Vemurafenib and Encorafenib had no impact on cell viability across all concentrations tested. Higher concentrations of Dabrafenib led to a statistically significant decrease in calcein-positive cells, however, viability did not drop below 90%. The next-generation inhibitor PLX8394 decreased cell viability after 6 h, but not after 1 h. Of note, effects of PLX8394 on the endothelial barrier were visible already after 1 h of treatment (Figure 4). In summary, the viability of cells treated with BRAFi did not correlate with their effects on barrier function.

→ We agree with the reviewer that adding the viability data strengthens our claims. We have added LIVE/DEAD assay data as Supplementary Figure S3 and have described it in the Results section (page 8 of the manuscript with markup).

Furthermore, isolation of skin endothelial cells is at risk to be accompanied with lymphatic endothelial cell contamination. The authors should provide data ensuring that the cells are of >90% endothelial cell purity by checking for PROX1-positive cells together with endothelial cells markers (CD31, VE-cadherin, uptake of AcLDL etc).

We acknowledge the reviewer's concerns about lymphatic endothelial cell (LEC) contamination. We were also concerned, that the presence of LEC might influence treatment effects and measurement outcomes. Upon closer inspection during our studies, however, we observed that the effects were independent of the endothelial cell type. To substantiate this finding, we performed additional ECIS measurements from three DMEC donors after sorting them into BEC and LEC (Supplementary Figure S5). The effect of relevant concentrations of BRAFi had comparable effects on endothelial

cells of lymphatic and blood endothelial origin. Especially the drastic barrier-disruptive effects of Vemurafenib were similar between BEC and LEC. Although testing for statistical significance also revealed p-values below a threshold of 0.05 for other treatments compared to their respective controls, the effect size in these cases was lower than 200 ohm in all cases, whereas Vemurafenib induced a drop of more than 1000 ohm. These data are now mentioned on page 8 of the revised manuscript. In additional immunofluorescence experiments of cultured DMEC, we confirmed effects of junction disruption upon BRAFi treatment in both Prox1-positive LEC and Prox1-negative BEC (Supplementary Figures S6 & S7). Furthermore, we want to note that in our analysis of patient biopsies, the loss of endothelial VE-Cadherin and Claudin-5 also occurred in Podoplanin-positive lymphatic vessels, providing additional evidence that the discussed effects of BRAFi are similar in BEC and LEC.

→ In summary, although BEC and LEC represent different vascular beds, we have shown that the detrimental effect of Vemurafenib is similar in both endothelial cell types as now seen in Supplementary Figures S5, S6 and S7. Additional text has been added to the manuscript on pages 8 and 14.

The work is of importance in understanding consequences for endothelial cells exposed to BRAF inhibitors used in the clinic using clinically relevant concentrations of the drugs investigated in vitro. If the authors provide with a major revision, the work could be acceptable for publication.

1) The Western blots in this manuscript are in general overexposed (saturated) and therefore differences between treatment conditions are not possible to be clearly defined. Therefore, quantifications of the experiments should be done and combined with representative Western blots.

We would like to emphasize that we used fluorescence-based detection of the molecular markers in our Western blot membranes, which has a higher dynamic range than peroxidase-based staining techniques. All blots are shown within a range, that allows for visible membrane background and bands below saturation. Furthermore, we acknowledge that the description of Figure 1 was too ambiguous, as there seems to have been a misunderstanding. The graph in figure 1C represents the quantification of the exact blots that are shown in Figure 1A-B (n = 1) for better direct comparison, whereas the graphs in Figure 1D show quantifications of the experiments in multiple blots (n = 3-4), of which one representative example is shown above. We apologize for the confusion and have denoted this more clearly in the figure legend of Figure 1.

→ Figure 1 and its figure legend now clearly show the quantifications of western blots. Together with the additional adjustments discussed in the following paragraphs we have improved data presentation and descriptions of Western blots in our manuscript.

2) Figure 1A-C n=?, notice no standard deviations in 1C. Does this mean that in 1C n=1?

Indeed, the graph in Figure 1C depicts the ratio of phosphorylated to total ERK1/2 in the exact blots that are shown in Figure 1A-B. This means n = 1 and it explains why there are no standard deviations. Since it is already known from literature that BRAFi can induce paradoxical activation of MAPK signaling in melanoma cells, the purpose of Figure 1A-C is to reiterate that this effect indeed happened in our primary NRAS-mutant melanoma cells and to provide a comparison to the effect in endothelial cells. The quantification and statistical analysis of paradoxical ERK1/2 activation in the

endothelium, which is the novel information, is then presented as quantifications of 3-4 experiments in Figure 1D in more detail.

→ Figure legends now clearly state the number of independent experiments in the respective panels.

3) Figure 1D, no significant differences? If there is no significant difference, then there is no difference between the treatments.

We thank the reviewer for this comment and have included the significance levels for the respective statistical tests in the graphs of Figure 1D.

4) Regarding concentrations of Vemurafenib; it is needed that the authors define endothelial cell viability (proliferation, Caspase-9 staining or LIVE/DEAD fixative stains) at this high concentration. Then 10 μ M is probably a too low concentration (see data in figure 2 where 10 μ M gives no data of relevance). Cell toxic effects could be the reason of increased passage of N-Fluorescein upon 100 μ M Vemurafenib treatment or the cause of cell-cell gaps (Figure 5). If 100 μ M truly shows that the cells are viable without any signs of toxicity, the paper would be more clear if main figures contain only 100 μ M Vemurafenib. It is recommended that cell cytotoxicity is tested for all compounds in this short- and long-term treatments

As described above, we agree with the reviewer that additional viability data would strengthen the manuscript. Therefore, we performed LIVE/DEAD stains of DMEC treated with respective concentrations of BRAFi for all timepoints used in functional assays.

→ We now show that the BRAFi-induced effects described in this manuscript are not due to cytotoxicity (Supplementary Figure S3). The results of this assay are mentioned in the Results section of the manuscript on page 8.

1) Figure 5; the authors should demonstrate the effect of BRAF inhibitors using a different approach. Trans-endothelial migration (trans-well), or similar methods would enforce the main message. Furthermore, migration defects could be evaluated by scratch-wound assay.

We assume the reviewer is referring to Figure 5B-C. We want to clarify that the aim of this assay was not to measure the migration of endothelial cells – in which case, we would agree that other methods would be more suitable. However, the main purpose of this assay was to measure visible differences in endothelial barrier function against invading tumor cells as a simplified model for metastasis formation. The tumor spheroids gradually displaced endothelial cells upon contact and formed gaps in the fluorescently labelled monolayer (as seen in Figure 5C). Previously, Obenauf et al had shown increased metastasis occurrence in mice treated with Vemurafenib and indeed also pre-treatment of human endothelial cells with 100 μ M Vemurafenib in our experiments facilitated the displacement of endothelial cells by tumor spheroids. We therefore conclude that the endothelial barrier is weaker and see this as a confirmation of Figure 5A and previous functional assays in Figure 4.

→ We apologize for the misunderstanding and have adjusted the schematic display of the assay and the legend of Figure 5.

Comment: the imaging in figure 5A is not clear enough to truly show the cell morphology and to define the cell status (see point 4 related to cell viability). We also advice that figure 5A also contains stainings for all other treatment conditions (or included in a supplement figure). What's the mechanism behind junctional rearrangement? Internalization, degradation or actin cytoskeleton-dependent mechanisms? Figure 5A, stainings should be quantified.

We agree, that the immunofluorescence images in Figure 5A do not suffice as a substitute for viability assays. As described above, LIVE/DEAD assays were performed to provide proof that cell viability is not affected by the respective treatments (Supplementary Figure S3).

We also thank the reviewer for the insightful comment regarding the dynamics of junctional rearrangement. Figure 5A now depicts changes in endothelial cell-cell junction architecture as well as the actin cytoskeleton. Upon treatment with 100 μ M Vemurafenib, many of the disrupted junction fragments were attached to actin stress fibers, indicating that there might be a cytoskeletal role in BRAFi-induced barrier dysfunction, similar to known events after treatment with other barrier-disruptive agents such as histamine, thrombin or VEGF (Gavard & Gutkind, 2008; Argaw *et al*, 2009).

As proposed by the reviewer, we also performed fluorescence signal quantification and indeed we saw a signal decrease of junction proteins upon Vemurafenib treatment.

→ Images of all inhibitors in three concentrations each are now shown in Supplementary Figures S6 (Vemurafenib, Dabrafenib) and S7 (Encorafenib, PLX8394). Image quantifications are now seen in Figure 5A as well as S6 and S7.

Figure 5C; with three asterisks in the figure, what is the actual significance and is it compared to DMSO? With the large SD the significance can hardly fit with three asterisks (<0.001).

We understand the reviewers concern about the significance level in Figure 5C (now called 5D), given the large standard deviation visible in the bar graph. To improve transparency and provide information about the distribution of measured data points, we have changed the graph to a violin plot and have included the individual measurement values as dots. In fact, 100 μ M Vemurafenib increased gap size with $p < 0.0001$ (****).

→ Quantification of measurements for Figure 5C are now depicted and described in detail as separate panel Figure 5D. Significant changes upon treatment are now marked with respect to the DMSO control treated endothelial monolayers.

5) Valuable skin biopsies of patients before and after treatment have been used for figure 6. The authors should pay more careful attention to what vasculature they are investigating in the biopsy material. The authors mainly focus on large arteries (large vascular lumens with a thick layer of ASMA-positive cells). We recommend that they investigate capillaries (5-10 μ m in diameter) which are more plastic and susceptible towards treatment.

We thank the reviewer for recognizing the value of patient biopsies in the context of this study. The effects of Vemurafenib were visible in all vessel types, yet the prominent features of arterial vessels in the images focused the reader's attention on this vessel type specifically. We selected representative images in Figure 6 to show multiple vessel types in each treatment, including larger arteries (identifiable by the thick layers of α -SMA), small capillaries (no α -SMA, but clear expression of VE-Cadherin and Claudin-5), and lymphatic vessels (positive for Podoplanin). The quantification of junction markers was performed in the peritumoral area of human melanoma samples among

all VE-Cadherin positive vessels including capillaries and separately in Podoplanin-positive lymphatic vessels (Table 2).

→ We have added a labeling system to the overlay images in Figure 6 to highlight different vessel types: arteries (A), capillaries (C) and lymphatic vessels (L).

Claudin-5 is a vascular marker but the antibody chosen clearly provides with high autofluorescence stains detecting blood cells in the vascular lumen and not only the endothelial cells. We therefore recommend to use another claudin-5 antibody that will stain dermal vasculature better.

We acknowledge the reviewer's concerns regarding autofluorescence and antibody specificity. Indeed, erythrocytes are visible in the channel we used for Claudin-5 staining. However, they are also visible in all other channels, which is due to their high autofluorescence – an inherent feature we cannot avoid. Since we want to enable the reader to distinguish more clearly between antibody staining and autofluorescence, we have included images of the respective samples showing the Sample Autofluorescence Channel (measured during scans with the Vectra Polaris imaging system). Importantly, the erythrocytes are located in the vessel lumen, whereas the true Claudin-5 signal is mainly overlapping with VE-Cadherin at the *tunica intima* of the vessel. To further illustrate specificity of the Claudin-5 signal, we have included images of an IgG control of the treatment-naive tissue sample in Figure 6. Upon close inspection, we conclude that the background visible in the stained sections is not due to an unspecific antibody, but due to high tissue autofluorescence, which is very common in skin, especially when using imaging channels with shorter wavelengths, as we did for α -SMA (FITC: 488 nm) and Claudin-5 (AF546: 546 nm). Importantly, we only quantified Claudin-5 signal in areas that were positive for VE-Cadherin or Podoplanin (Table 2), to ensure we only measured signals in the endothelial layer.

→ We have included images for sample autofluorescence and IgG controls in Figure 6 and have specified the quantified area in Table 2.

Which patient is imaged in figure 6?

Figure 6 depicts representative images from patient #1, before and during Vemurafenib monotherapy.

→ We have now specified this in the figure legend and apologize for the missing information.

Please prepare a supplement figure with patient 1-4 to show representative images of the main differences.

→ In agreement with the reviewer's request, we have now added Supplementary Figure S8, which depicts representative images from patients #2-5, that are not shown in the main figure.

Do the authors expect that Vemurafenib 100 μ M will also decrease VE-cadherin and claudin-5 total protein levels?

The reviewer posed an interesting question. In our patient samples, the signal intensity of vascular markers VE-Cadherin and Claudin-5 was decreased upon Vemurafenib treatment, which suggested a loss of junctional epitopes in patient vessels (Figure 6). Our proteomics data did not show a substantial loss of total protein upon Vemurafenib treatment, as seen in Figure S2. However, immunofluorescence images displayed a small decrease in signal intensity of VE-Cadherin and Claudin-5 in DMEC treated with

Vemurafenib (Figure 5A). It is worth noting, that we used relatively short timepoints in cell culture (1 – 6 h), whereas patients had received BRAFi therapy for longer periods before biopsies were taken. Given the fact that availability of patient tissue in our retrospective study was limited to dermal metastases excised for medical reasons, we can only hypothesize that Vemurafenib therapy will lead to loss of junction proteins, and therefore an impairment of vascular integrity over time.

→ Image quantifications have been added to Figure 5A.

6) Table 2: The quantification is not clear. The authors should describe the data in a more descriptive way. For example, what does it mean to have more than 100% (181.41% of claudin-5 for patient 5) of the vascular markers? Also, it is not realistic to describe percentage data with 2 decimals. The authors should also classify their quantification based on vessel type (large caliber vessels vs capillaries), cancer and pseudo-normal tissue.

We apologize for not sufficiently explaining the quantification method in the manuscript. Results are shown as fluorescence intensity of the respective marker in percent of their fluorescence intensity in treatment-naive tissues. Of note, the Patient #5 sample during treatment, that generally presented with higher fluorescence intensity measurements across all channels, was matched with a control sample from a different patient. Furthermore, we want to emphasize that the quantification of vascular markers (Table 2) was performed in larger areas of the tissue sections, focusing on the tumor-adjacent skin and excluding the tumor area itself. We also did, in fact, distinguish between vessel types by quantifying the signal intensity of junction markers within Podoplanin-positive areas (lymphatic vessels) and all VE-Cadherin-positive vessels. Unfortunately, the independent quantification of vessels positive for α -SMA was not possible with our method, because the α -SMA signal in smooth muscle did not overlap with junctional markers in endothelial cells. However, independent analysis of the samples with further tools can be performed once the images are published. Table 2 and Figure 6 demonstrate that the loss of junctional markers upon Vemurafenib therapy is detectable in all vessel types.

→ We apologize for not providing a clearer description of results, and we have now added more detailed information on the quantification and effects on different vessel types in the Results section (page 9).

As a way to validate their in vitro findings (permeability and junctional disruption in these patient tissue biopsies), the authors should check for leakage by staining for serum proteins like IgG, fibrinogen or serum albumin.

We thank the reviewer for this suggestion and performed a staining of the biopsy material for fibrinogen. Indeed, we could generally identify accumulation of fibrinogen in the interstitial space surrounding blood vessels, as can be seen in Figure R1 below. However, we encountered several difficulties regarding the quantification of the fibrinogen signal. Our previous analysis method was not suitable since it depended on measuring signal intensities only within areas that were positive for a vascular marker such as VE-Cadherin. Quantification of fibrinogen in the perivascular space would therefore require the manual selection of ROIs, which would introduce bias, especially given the low number of patients. Figure R1B below also shows areas of tissue hemorrhage, potentially associated with the excision process of the biopsies, which could further confound quantification results. Therefore, we refrain from including this part in the manuscript, but instead show exemplary images from the fibrinogen staining below for the reviewer and for interested readers (Figures R1 & R2).

Figure R1: Immunofluorescence images of biopsy material from patient #1 before (naive) and during Vemurafenib monotherapy. The biopsy material was stained as stated in the methods section of the main manuscript, with the addition of a FITC-labelled anti-Fibrinogen antibody (1:60, #F0111, Dako, Glostrup, Denmark). A: Exemplary images of blood and lymphatic vessels show perivascular fibrinogen accumulation (arrows). Scale bars = 50 μ m. B: Exemplary images of tissue hemorrhage in biopsies from patient #1 show areas with high numbers of erythrocytes (high autofluorescence) and fibrinogen. Scale bars: 100 μ m.

Figure R2: Immunofluorescence of biopsy material of patient #2 - #5 before (naive) and during the indicated therapy. Tissues were stained according to Figure R1 above. V + C = Vemurafenib + Cobimetinib, D + T = Dabrafenib + Trametinib. Scale bars = 50 μ m.

Minor comments: important issues that can confidently be addressed:

The authors want to fill a gap in knowledge related to BRAF inhibitors effect on endothelial cells, which a limited number of publications are available.

2) Why are the authors using CellTracker for visualize cell morphology. It would be better if cells were stained for VE-cadherin and beta-actin including nuclear stain with DAPI. This would far better define the cell morphology after treatments.

→ We thank reviewer #1 for the input and performed a staining of the mentioned junctional and cytoskeletal markers (Figure 5A). Additionally, we have added images of a Prox1 staining to compare the effects on blood and lymphatic endothelial cells (Supplementary Figures S6 and S7).

3) Please in Material & Methods describe KinSwing activity predictions index to help the reader to follow the results better.

→ We appreciate this comment and have added a brief explanation of the KinSwing scoring system in the Methods section (page 20) for better understanding.

4) Table 1 could be reformatted to be more easily to read.

→ We have reformatted Table 1 for readability. Kinases are now sorted according to the number of directly binding inhibitors, which facilitates the comparison among inhibitors. Additionally, we have omitted the last three rows, which summarize the number of identified kinases, as this information is already described in the main text (page 7-8).

5) Figure 1, is ERK= ERK1/2?

Indeed, we used antibodies to stain for ERK1/2. We thank the reviewer for pointing out the lack of clarity here.

→ We have adjusted the labels in Figure 1 and S1 accordingly and also specified the staining for ERK1/2 and their respective phosphosites in the figure legends.

6) The discussion text should be shortened and more focused towards their findings and with conclusions of performed experiments. How is the paradoxical effect of Vemurafenib (figure 1) related to their later findings (Figure 2 and 3)? In other words, what is the relation between figure 1 and figure 2 and 3?

Even though paradoxical MAPK activation has been widely discussed as a possible reason for adverse events, our study provides proof that this is not the case for endothelial cells, and that other off-target mechanisms are responsible for functional defects in this cell type.

→ We have made deletions on pages 13 and 14 to make the discussion part more concise and have focused on the information that is relevant to interpret our results in Figures 1, 2 and 3. We have also added a sentence at the beginning of the part “Off-targets in endothelial cells” on page 11 of the manuscript to connect our data on paradoxical MAPK activation with later results.

7) For the discussion; is the result in figure 5C supported by data that patients on Vemurafenib treatment would be exposed to a higher risk of metastasis?

To the best of our knowledge there is no published study or registered prospective trial for addressing this question. Current published papers focus on the long-term clinical efficacy of Vemurafenib in melanoma patients, such as:

- The long-term follow up data of the pivotal study comparing Dacarbazine versus Vemurafenib showed that Vemurafenib led to superior OS and PFS compared to Dacarbazine (McArthur *et al*, 2014).
- As part of a real-life long-term follow up study on Vemurafenib, metastases at new sites were detected. However, it was further argued, that Vemurafenib monotherapy should be continued if there is only a limited progression (Puzanov *et al*, 2015).

In summary, these reports do not allow us to draw conclusions on a potentially higher risk of resistant melanoma and higher risk of metastases in patients treated with Vemurafenib monotherapy. This would need a clinical trial or more detailed retrospective analyses of clinical trial data if available.

8) Figure 3B, resolution of text needs to be improved and the full compound names could be written in figure 3A.

→ We appreciate the reviewer's comment and have adapted Figure 3 accordingly.

9) Figure 4A are any of the results statistically significant? If not, then there is no difference.

We have adapted Figure 4A and included graphs that depict the barrier resistance change after 1 h of BRAFi treatment. We have performed statistical testing (one-way ANOVA with Dunnett's test for multiple comparisons) on these data and included the respective significance levels in the bar graphs. While higher concentrations of Vemurafenib and PLX8394 induced substantial drops in barrier resistance, only the highest dose of Dabrafenib led to a barrier decline. Encorafenib had no significant effect even at 100 μ M.

→ Figure 4A, the figure legend and the respective part in the Results section (page 8) have been adapted accordingly.

10) The authors should elaborate a hypothesis based on their phosphoproteomics data. Which of the off-targeted molecule(s) could impact endothelial barrier?

From our current results, we are not yet able to tell which of the off-target molecules is responsible for BRAFi-induced barrier disruption. We propose that candidates such as RHOA, CDKs, PKA, or GSK3- β and interactions between them should be investigated further, as they were differentially regulated in our phosphoproteomics dataset and are known from the literature in the field as important signaling molecules for vascular permeability.

→ We have included a brief statement regarding the hypotheses we generated from comparing the phosphoproteomics and KinSwing data with the functional effects at the end of the Discussion part termed "Functional implications" (page 14).

****Referees cross-commenting****

With our deep knowledge in endothelial cell biology, we would like to emphasize the need of Bromberger *et al* to reply to our comments. Additional experiments and verifications will improve the impact of the performed research. With reviewer 2 demanding far less additional work to be done there is a discrepancy between the two reviewers of the estimated time needed for performing a revision (1-3 months for reviewer 1 versus 1 month for reviewer 2). I

(reviewer 1) believe that at least three (3) moths will be needed to collect additional data to reply to the questions.

Reviewer #1 (Significance (Required)):

SIGNIFICANCE:

This manuscript addresses an important question; how is the vasculature affected by cancer treatments? It is not unusual that the vascular status is neglected in clinical treatment studies. The manuscript provides valuable phosphoproteomics data of great interest related to this topic. The major weakness of the work is the lack of data verifying the chosen concentrations for the BRAF inhibitors used in the study. There is a great risk that several results based on the 100 uM Vemurafenib treatment (of high impact for the story) are based on cell toxicity due to a high concentration treatment in vitro. Also, the link between the strategy of performed in vitro experiments isn't clear and there is a lack of connecting the in vitro data to the validation performed on melanoma patient tissue biopsies. It is a great strategy to investigate skin biopsies before and after treatment. The precious biopsy material should be more carefully investigated and evaluated.

We thank the reviewer for their insightful comments and have incorporated additional viability data in the manuscript, confirming the presence of intact monolayers of viable endothelial cells following treatment with the respective inhibitors. Indeed, in our cell culture data, we showed that endothelial barrier function was decreased upon treatment with Vemurafenib and PLX8394. This was confirmed by ECIS, transwell assays, spheroid invasion assays and immunofluorescence data. We now also show that upon treatment with Vemurafenib endothelial junction markers VE-cadherin and Claudin-5 were decreased in human endothelial cells in vitro and in tissue from a melanoma patient.

- ➔ As pointed out by the reviewer, we now show in Supplementary Figure S3 that the effects described in this manuscript are not due to cytotoxicity. Furthermore, we have added additional images from biopsy material and provided clarifications in the revised manuscript.

AUDIENCE: after improvement of the manuscript by better presentation of existing data and by additional experiments the work presented would be of interest to a pre-clinical and clinical audience investigating cancer treatments.

With this manuscript, that includes data from primary human endothelial cells and human biopsies, we hope to provide valuable insights for the medical research community. We thank the reviewer for finding our results of interest to pre-clinical and clinical audience.

References

- Argaw AT, Gurfein BT, Zhang Y, Zameer A & John GR (2009) VEGF-mediated disruption of endothelial CLN-5 promotes blood-brain barrier breakdown. *Proc Natl Acad Sci U S A* 106: 1977–1982
- Gavard J & Gutkind JS (2008) Protein Kinase C-related Kinase and ROCK Are Required for Thrombin-induced Endothelial Cell Permeability Downstream from G α 12/13 and G α 11/q. *J Biol Chem* 283: 29888–29896
- McArthur GA, Chapman PB, Robert C, Larkin J, Haanen JB, Dummer R, Ribas A, Hogg D, Hamid O, Ascierto PA, *et al* (2014) Safety and efficacy of vemurafenib in BRAFV600E and BRAFV600K mutation-positive melanoma (BRIM-3): Extended follow-up of a phase 3, randomised, open-label study. *Lancet Oncol* 15: 323–332
- Puzanov I, Amaravadi RK, McArthur GA, Flaherty KT, Chapman PB, Sosman JA, Ribas A, Shackleton M, Hwu P, Chmielowski B, *et al* (2015) Long-term outcome in BRAFV600E melanoma patients treated with vemurafenib: Patterns of disease progression and clinical management of limited progression. *Eur J Cancer* 51: 1435–1443

Reviewer #2 (Evidence, reproducibility and clarity (Required)):

Summary:

The authors provide insight into the gaps within BRAFi research in an effort to further understand how elements such as mechanisms of resistance and clinically observed adverse events in melanoma patients occur. This manuscript more specifically highlights the effects of BRAFi treatment on endothelial cells in the context of vasculature. The authors begin to explore how traditional BRAFi therapies may lead to such adverse events due to the role they play alongside that of targeting melanoma cells such as off-target effects, paradoxical endothelial signaling, and inducing a pro-tumorigenic microenvironment. The conducted studies demonstrate simple and effective methodology, focusing on proteomic and phosphoproteomic analysis, to elucidate the endothelial consequences of BRAFi treatment. The authors provide sound conclusions from the presented data and validate their in vitro findings with clinical observations using patient tissue. The analysis within this manuscript is just scratching the surface and leaves the authors with much to explore in future manuscripts.

Major comments:

The authors provide a solid story outlining the pitfalls in BRAFi therapy research and the consequences on endothelial vasculature in the treatment of BRAF mutant melanoma. The manuscript details clinical relevance of the research, functional impact to the field, and a thorough discussion on the scope of this work and where it may be lacking, which allows for the opportunity for future directions.

We thank the reviewer for their appreciation of our work.

Minor comments:

The authors may consider revising minor errors within the Discussion as indicated below.

Discussion - Paradoxical MAPK activation

Missing comma between cells and the; "For endothelial cells, the concentration of BRAFi measured in the patient circulation is critical."

→ We have corrected this error in the Discussion section (page 10).

Discussion - Off targets in endothelial cells

Missing comma between range and it; "At concentrations in the low μM range, it inhibits numerous other kinases."

→ We thank the reviewer for this comment. The mentioned sentence has been rephrased in the revised version of the manuscript (page 11).

Missing commas around apart from MAPK; "This suggests that, apart from MAPK, other signaling pathways would also be affected by BRAFi treatment"

→ We have corrected the mentioned errors in the Discussion section of the revised manuscript (page 11).

Reviewer #2 (Significance (Required)):

This manuscript poses a key discussion in the importance of expanding research of molecular targeted therapies on more than just the target cells as the consequences to surrounding cell types can give vital insight into potential adverse effects in the clinic. The authors note that while this is not a novel concept, there are still gaps that prove vital in understanding clinical impact, which they hope to fill with this manuscript. They provide support to their conclusions using primarily proteomic approaches with the addition of some comparative analysis of a publicly available dataset, and patient tissue samples in order to validate their findings. Whether in the context of treating melanoma or any other disease. this manuscript serves as a helpful reminder to pre-clinical and clinical researchers alike in how important it is to factor in the patient as a whole, not just the disease when identifying effective treatment options.

May 6, 2024

RE: Life Science Alliance Manuscript #LSA-2024-02671-TR

Dr. Klaudia Schossleitner
Medical University of Vienna
Department of Dermatology
Währinger Gürtel 18-20
Vienna 1090
Austria

Dear Dr. Schossleitner,

Thank you for submitting your revised manuscript entitled "Off-targets of BRAF inhibitors disrupt endothelial signaling and vascular barrier function". We would be happy to publish your paper in Life Science Alliance pending final revisions necessary to meet our formatting guidelines.

- please be sure that the authorship listing and order is correct
- please update the Data Availability statement to include accession info
- the Supplementary Methods should be incorporated into the main Materials and Methods section
- please move the Supplemental figure legends after the main figure legends in the main manuscript file, rather than in a separate file
- since Figure S1 only has one panel, please remove the "A" label in the figure and legend
- please add callouts for Figure 6A and B
- please add the Twitter handle of your host institute/organization as well as your own or/and one of the authors in our system

Figure Checks

- please add sizes next to all blots

A. FINAL FILES:

B. MANUSCRIPT ORGANIZATION AND FORMATTING:

Sincerely,

May 20, 2024

RE: Life Science Alliance Manuscript #LSA-2024-02671-TRR

Dr. Klaudia Schossleitner
Medical University of Vienna
Department of Dermatology
Währinger Gürtel 18-20
Vienna 1090
Austria

Dear Dr. Schossleitner,

Thank you for submitting your Research Article entitled "Off-targets of BRAF inhibitors disrupt endothelial signaling and vascular barrier function". It is a pleasure to let you know that your manuscript is now accepted for publication in Life Science Alliance. Congratulations on this interesting work.

DISTRIBUTION OF MATERIALS:

Again, congratulations on a very nice paper. I hope you found the review process to be constructive and are pleased with how the manuscript was handled editorially. We look forward to future exciting submissions from your lab.

Sincerely,
